# Large fresh water influx induced salinity gradient and diagenetic changes in the northern Indian Ocean dominate the stable oxygen isotopic variation in *Globigerinoides ruber*

Rajeev Saraswat[1]*, Thejasino Suokhrie[1], Dinesh K. Naik[2], Dharmendra P. Singh[3], Syed M. Saalim[4], Mohd Salman[1,5], Gavendra Kumar[1,5], Sudhira R. Bhadra[1], Mahyar Mohtadi[6], Sujata R. Kurtarkar[1], Abhayanand S. Maurya[3]

[1] Micropaleontology Laboratory, National Institute of Oceanography, Goa, India

[2] Banaras Hindu University, Varanasi, Uttar Pradesh, India

[3] Indian Institute of Technology, Roorkee, India

[4] National Center for Polar and Ocean Research, Goa, India

[5] School of Earth, Ocean and Atmospheric Sciences, Goa University, Goa

[6] MARUM, University of Bremen, Bremen, Germany

*Correspondence to*: Rajeev Saraswat (rsaraswat@nio.org)

**Abstract.** The application of stable oxygen isotopic ratio of surface dwelling planktic foraminifera *Globigerinoides ruber* (white variety) ($\delta^{18}O_{ruber}$) to reconstruct past hydrological changes requires precise understanding of the effect of ambient parameters on $\delta^{18}O_{ruber}$. The northern Indian Ocean, with huge freshwater influx and being a part of the Indo-Pacific Warm Pool, provides a unique setting to understand the effect of both the salinity and temperature on $\delta^{18}O_{ruber}$. Here, we use a total of 400 surface samples (252 from this work and 148 from previous studies), covering the entire salinity end member region, to assess the effect of fresh water influx induced seawater salinity and temperature on $\delta^{18}O_{ruber}$ in the northern Indian Ocean. The analyzed surface $\delta^{18}O_{ruber}$ very well mimics the expected $\delta^{18}O$ calcite estimated from the modern seawater parameters (temperature, salinity and seawater $\delta^{18}O$). We report a large diagenetic overprinting of $\delta^{18}O_{ruber}$ in the surface sediments with an increase of 0.18‰ per kilometer increase in water depth. The fresh water influx induced salinity exerts the major control on $\delta^{18}O_{ruber}$ ($R^2 = 0.63$) in the northern Indian Ocean, with an increase of 0.29‰ per unit increase in salinity. The relationship between temperature and salinity corrected $\delta^{18}O_{ruber}$ ($\delta^{18}O_{ruber} - \delta^{18}O_{sw}$) in the northern Indian Ocean [T= -0.59*($\delta^{18}O_{ruber} - \delta^{18}O_{sw}$) + 26.40] is different than reported previously, based on the global compilation of plankton tow $\delta^{18}O_{ruber}$ data. The revised equations will help in better paleoclimatic reconstruction from the northern Indian Ocean by using the stable oxygen isotopic ratio.

## 1. Introduction

The stable oxygen isotopic ratio ($\delta^{18}O$) of biogenic carbonates is one of the most extensively used marine paleoclimatic proxies (Mulitza et al., 1997; Lea, 2014; Metcalfe et al., 2019; Saraswat et al., 2019). Even though it was initially suggested that the oxygen isotopic fractionation in biogenic carbonates is largely driven by temperature (Urey et al., 1947), subsequent work revealed that besides temperature, salinity and carbonate ion concentration of ambient seawater also affects the biogenic carbonate $\delta^{18}O$ (Vergnaud-Grazzini, 1976; Spero et al., 1997; Bemis et al., 1998; Spero et al., 1997; Bijma et al., 1999; Mulitza et al., 2003). On longer time-scales, the global ice volume contributes the largest fraction (~1.0-1.2‰) of the glacial-interglacial shift in marine biogenic carbonate $\delta^{18}O$, at a majority of the locations (Shackleton, 1987; 2000; Lambeck et al., 2014). The ice volume changes have induced well-defined shifts in biogenic carbonate $\delta^{18}O$ during the last several million years. Therefore, the regional evaporation-precipitation, runoff and temperature changes are reconstructed from the global ice-volume corrected biogenic carbonate $\delta^{18}O$ (Wang et al., 1995; Kallel et al., 1997; Schmidt et al., 2004; Saraswat et al., 2012; 2013; Kessarkar et al., 2013).

The $\delta^{18}O$ of surface dwelling planktic foraminifera *Globigerinoides ruber* ($\delta^{18}O_{ruber}$) is often used to reconstruct past surface seawater conditions (Saraswat et al., 2012; 2013; Mahesh and Banakar, 2014). Therefore, continuous efforts are made to understand the factors affecting $\delta^{18}O_{ruber}$ (Vergnaud-Grazzini, 1976; Mulitza et al., 1997; 2003; Waelbroeck et al., 2005; Mohtadi et al., 2011; Horikawa et al, 2015; Hollstein et al., 2017; Sanchez et al., 2022). The depth habitat of *G. ruber* in the tropical Atlantic Ocean has been inferred from its stable oxygen isotopic ratio (Farmer et al., 2007). The change in stable oxygen isotopic ratio of planktic foraminifera, including *G. ruber*, is suggested as a proxy to reconstruct upper water column stratification in the tropical Atlantic Ocean, based on the good correlation between $\delta^{18}O$ and the ambient seawater characteristics (Steph et al., 2009). A few studies suggested a difference in the $\delta^{18}O$ of various morphotypes of *G. ruber* (sensu stricto and sensu lato) and attributed it to their distinct ecology and depth habitat (Löwemark et al., 2005). However, a recent study from the Gulf of Mexico suggested a similar ecology and depth habitat for both the *G. ruber* morphotypes (Thirumalai et al., 2014). The northern Indian Ocean being influenced by huge fresh water influx as well as being a part of the Indo-Pacific Warm Pool (De Deckker, 2016), provides a unique setting to understand the effect of large salinity and temperature changes on $\delta^{18}O_{ruber}$. Earlier, Duplessy et al., (1981) measured $\delta^{18}O$ of the living *G. ruber* specimens collected from the water column as well as of the dead ones recovered from surface sediments of the northern Indian Ocean. A similar study from the Red Sea and adjoining western Arabian Sea suggested that *G. ruber* calcifies its test in isotopic equilibrium with the ambient seawater, thus tracking the inter-annual subtle change in the salinity and temperature (Kroon and Ganssen, 1989; Ganssen and Kroon, 1991).

The temperature influence on $\delta^{18}O_{ruber}$ is well defined (Multiza et al., 2003). The effect of fresh water influx induced changes in ambient salinity on $\delta^{18}O_{ruber}$ is, however, debated (Dämmer et al., 2020). With the extensive use of $\delta^{18}O_{ruber}$ to reconstruct regional evaporation-precipitation changes, especially from the monsoon dominated tropical oceans, it is imperative to understand the precise influence of ambient salinity on $\delta^{18}O_{ruber}$. The ambient seawater pH, carbonate ion concentration (Bijma et al., 1999), presence/absence of symbionts (Jørgensen et al., 1985) also affect the isotopic composition of *G. ruber*. However, a limited glacial-interglacial variability in these parameters is masked

by the dominance of temperature and fresh water influx induced salinity changes in the oxygen isotopic ratio of *G.*
*ruber.* Additionally, the diagenetic changes, especially dissolution, also substantially alters the original isotopic
composition of the foraminifera shells (Berger and Killingley, 1977; Wu and Berger, 1989; Lohmann, 1995; McCorkle
et al., 1997; Wycech et al., 2018). The dissolution preferentially removes lighter oxygen rich sections of the shells,
thus increasing the whole shell $\delta^{18}O_{ruber}$ (Berger and Gardner, 1975; Lohmann, 1995; Weinkauf et al., 2020). The
studies based on the comparison of ambient parameters with the isotopic composition of living specimens collected
in plankton tows may not address the complete range of the changes in the isotopic signatures during the sinking of
the tests from the surface waters post death, and its subsequent deposition in the sediments at the bottom of the sea.
As the fossil shells are the sole basis to find out the isotopic ratio of the ambient seawater in the past, the effect of
diagenetic changes including the dissolution on foraminifer's oxygen isotopic ratio has to be properly evaluated. Here,
we assess the influence of fresh water influx induced strong salinity gradient, ambient temperature, depth induced
dissolution and other associated parameters on the stable oxygen isotopic ratio of the surface dwelling planktic
foraminifera *G. ruber* (white variety) in the surface sediments of the northern Indian Ocean.

## 2.  Ecology of *Globigerinoides ruber* (white)

*Globigerinoides ruber* is a spinose planktic foraminifera inhabiting the mixed layer waters, throughout the year, in the
tropical-subtropical regions (Guptha et al., 1997; Kemle-von-Mücke and Hemleben, 1999). It is one of the dominant
planktic foraminifera in the northern Indian Ocean (Bé and Hutson, 1977; Bhadra and Saraswat, 2021) with its relative
abundance being as high as ~60% (Fraile et al., 2008). Its test is medium to low trochospiral and hosts algal symbionts
(Hemleben et al., 1989). *Globigerinoides ruber* prefers to feed upon phytoplankton (Hemleben et al., 1989), and is
dominant in oligotrophic warmer water with optimal temperature being 23.5°C (Fraile et al., 2008). However, it is
amongst a few planktic foraminifera species that can tolerate a wide range of salinity (22-49 psu) and temperature
(14-31°C) (Hemleben et al., 1989; Guptha et al., 1997). Two varieties of *G. ruber,* namely the white and pink are
common in the world oceans. However, the pink variety of *G. ruber* became extinct in the Indian and Pacific Oceans
at ~120 kyr during the Marine Isotopic Stage 5e (Thompson et al., 1979).

## 3.  Northern Indian Ocean

The Indian Ocean with its northern boundary in the tropics includes two hydrographically contrastingly basins, namely
the Arabian Sea and Bay of Bengal (BoB) (Figure 1). The excess of evaporation over precipitation generates high
salinity water mass that spreads throughout the surface of the northern Arabian Sea with its core as deep as ~100 m
(Shetye et al., 1994; Prasanna Kumar and Prasad, 1999; Joseph and Freeland, 2005). Other high salinity water masses
from both the Persian Gulf and Red Sea enter the northern Arabian Sea at deeper depths between 200-400 m and 500-
800 m, respectively (Rochford, 1964). A strong upwelling along the western boundary of the Arabian Sea during the
summer monsoon season brings cold, nutrient rich subsurface waters to the surface (Chatterjee et al., 2019). The weak
upwelling during the same season is also reported in the southeastern Arabian Sea (Smitha et al., 2014).

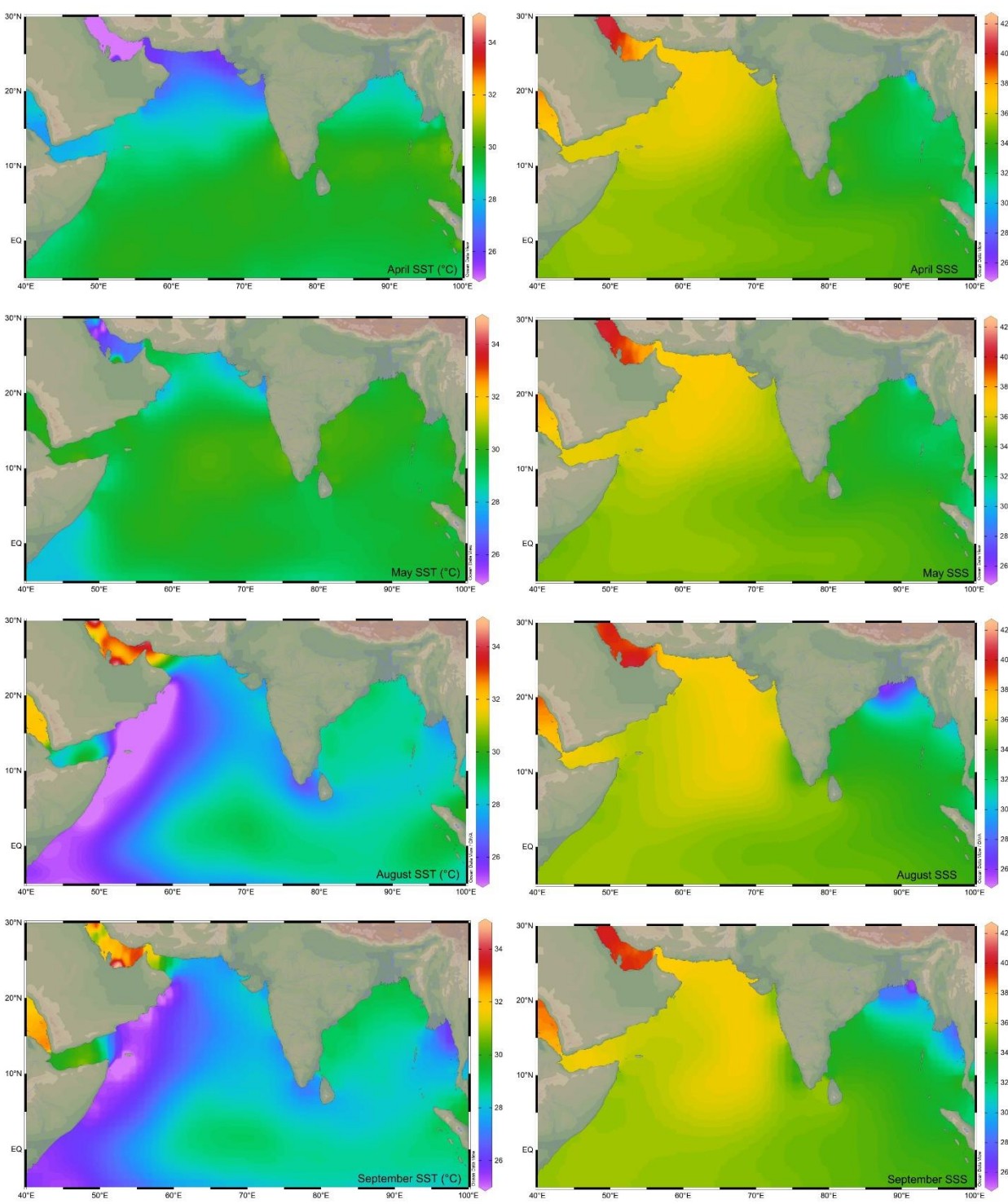


**Figure 1: The sea surface temperature (SST) (°C) (Locarnini et al., 2018) and salinity (SSS) (psu) (Zweng et al., 2018) in the**

**northern Indian Ocean during the monsoon (August-September) and non-monsoon (April-May) months. The major rivers**


The surface water is relatively fresher in the BoB, as the majority of the rivers from the Indian sub-continent

drain here, with the total annual continental runoff accounting to 2950 km$^3$ (Sengupta et al., 2006). Additionally, the
total annual precipitation over the BoB is 4700 km$^3$, and the evaporation is 3600 km$^3$ (Sengupta et al., 2006). The high
salinity Arabian Sea water is transported into the BoB and the fresher BoB water mixes with the high salinity Arabian
Sea water, by the seasonally reversing coastal currents (Shankar et al., 2002). The upwelling during summer is
restricted to only the northwestern part of the BoB (Shetye et al., 1991). The upwelling combined with the convective
mixing during the winter season in the north-eastern Arabian Sea (Madhupratap et al., 1996) as well as eddies in the
BoB (Prasanna Kumar et al., 2004; Sarma et al., 2020) result in very high primary productivity in both the basins
(Qasim, 1977; Prasanna Kumar et al., 2009). The high primary productivity and fresh water capping induce strong
stratification and restricted circulation, that creates oxygen deficient zones (ODZ) at the intermediate depth in both
the Arabian Sea (Rixen et al., 2020; Naqvi, 2021) and BoB (Bristow et al., 2016, Sridevi and Sarma, 2020). The
Arabian Sea ODZ, however, is comparatively thicker and intense, leading to denitrification (Naqvi et al., 2006), which
is not reported yet from the BoB (Bristow et al., 2016).

The equatorial Indian Ocean forms a part of the Indo-Pacific Warm Pool with the sea surface temperature

>28 °C throughout the year (Vinayachandran and Shetye, 1991; De Deckker, 2016). The marginal regions of the BoB
are comparatively warmer due to the fresh water influx from the rivers. The riverine influx shoals the mixed layer and
thickens the barrier layer, a buoyant layer separating the thermocline from the pycnocline, in the BoB (Howden and
Murtugudde, 2001). The riverine influx flows as a low salinity tongue all along the eastern margin of India (Chaitanya
et al., 2014). The annual average sea surface salinity (SSS) is <34 psu throughout the BoB, increasing from the head
bay towards south. In contrast to that, SSS remains >35 psu almost throughout the year in the Arabian Sea (Rao and
Sivakumar, 2003). The excess of evaporation over precipitation due to the dry northeasterly winds leads to the highest
salinity in the northern BoB during the winter (Rao and Sivakumar, 2003).

**4.  Materials and Methodology**
The surface sediments were collected all along the path of the seasonal coastal currents in the northern Indian Ocean
(Figure 2, Supplementary Table 1). The samples from the Ayeyarwady Delta Shelf in the northeastern BoB were
collected during 'India–Myanmar Joint Oceanographic Studies' onboard Ocean Research Vessel *Sagar Kanya*
(SK175). A total of 110 surface sediment samples were collected from the water depths ranging from 10 m to 1080
m, on the Ayeyarwady Delta Shelf (Ramaswamy et al., 2008). The multicore samples were also collected at regular
intervals in transects running perpendicular to the coast, from the western BoB during the cruise SK308, onboard
Research Vessel *Sindhu Sadhana* (cruise SSD067) and Research Vessel *Sindhu Sankalp* (cruise SSK35). A total of
84 surface samples (including 71 multicore samples and 13 grab samples from sandy sediments) were collected from
the inner shelf to outer slope region of the eastern margin of India during the cruise SK308 (Suokhrie et al., 2021a;
Saalim et al., 2022). These samples from the western BoB represent the lowest salinity region in the northern Indian
Ocean (Panchang and Nigam, 2012). The multicore samples collected between 25 m and 2980 m in the Gulf of Mannar
and the region west of it (43 samples onboard Research Vessel *Sindhu Sadhana* SSD004) represent the zone of cross-
basin exchange of seawater between the BoB and the Arabian Sea (Singh et al., 2021). The spade core samples
collected from the southeastern Arabian Sea (ORV *Sagar Kanya* cruise SK117 and SK237) are located close to the
distal end of the low salinity BoB water intruding into the Arabian Sea. The multicore samples (13 number) collected
during SSD055 cruise, from the northeastern Arabian Sea, represent the warm saline conditions. We also collected
spade core surface samples from the Andaman Sea, onboard Research Vessel *Sindhu Sankalp* (cruise SSK98). A total
of 252 samples had sufficient *G. ruber* for isotopic analysis (Table 1). The new data was augmented with 148
previously published core-top studies (e.g. Sirocko, 1989; Prell and Curry, 1981; Duplessy et al., 1981; 1982) mainly
compiled by Waelbroeck et al., (2005). Therefore, a total of 400 surface sample data points were used for this study.

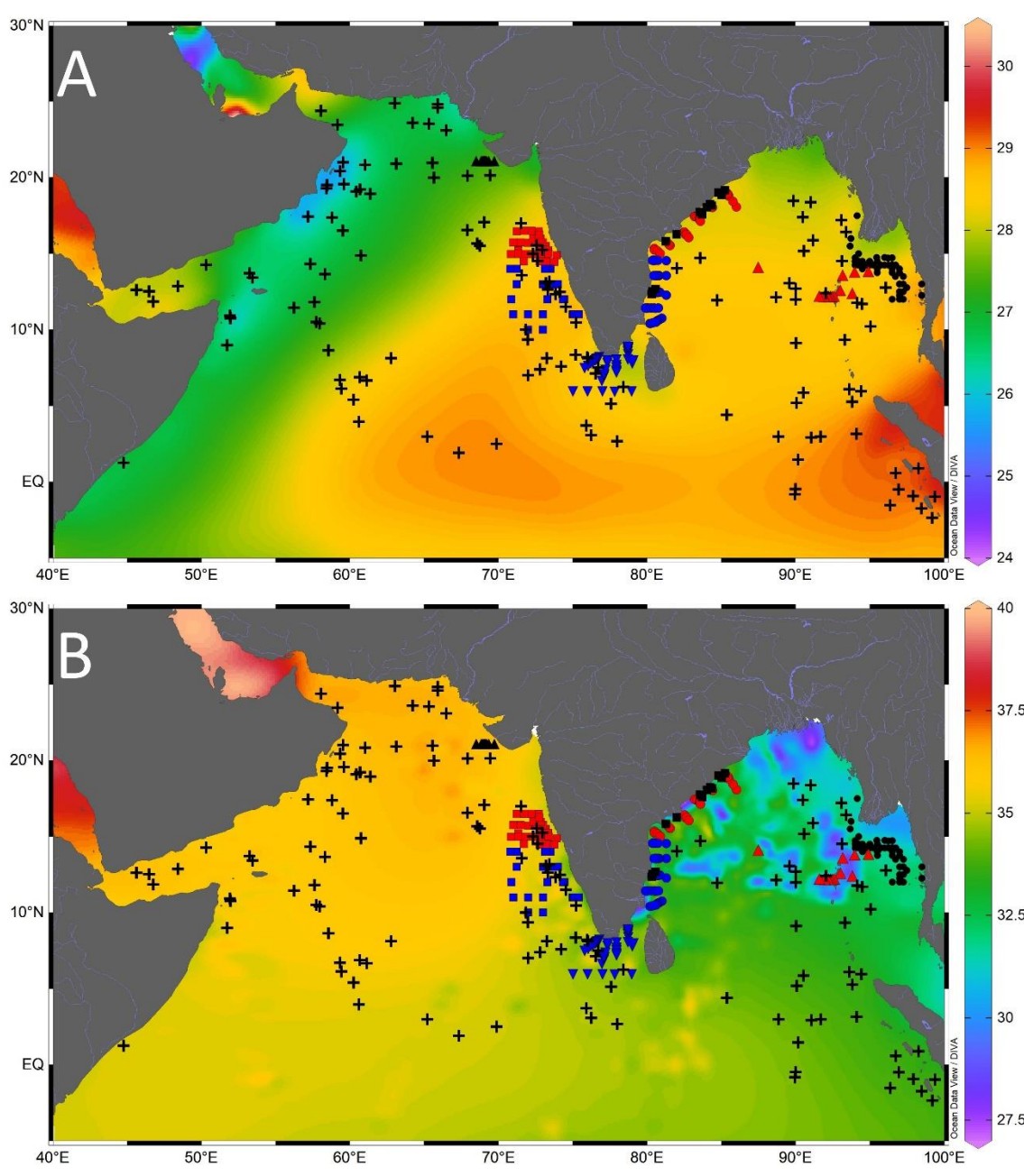

**Figure 2: Location of the core top samples analyzed in this study (black filled triangle - cruise SSD055, red filled square -**
**cruise SK117, blue filled square - cruise SK237, blue filled inverted triangle - cruise SSD004, blue filled circle - cruise**
**SSD067, red filled circle - cruise SK308, black filled square - cruise SSK035, red filled triangle - cruise SSK098, black filled**
**circle - cruise SK175), and the previously published core top values (black plus) compiled from the northern Indian Ocean.**
**The background contours are temperature (°C) (A) and salinity (psu) (B) with the scale on the right. Major rivers draining**
**into the northern Indian Ocean, are marked by thin blue lines. The map was prepared by using Ocean Data View software**
**(Schlitzer, 2018).**

**Table 1: Details of the timing, region and the number of surface sediment samples collected in each expedition, used in this study (SK-** *Sagar Kanya***, SSD-** *Sindhu Sadhana***, SSK-** *Sindhu Sankalp***).**

| Sr.No. | Cruise | Month/Year | Area | Total Samples |
|--------|--------|------------|------|---------------|
| 1. | SK117 | September-October 1996 | Eastern Arabian Sea | 27 |
| 2. | SK175 | April-May 2002 | North-eastern Bay of Bengal | 45 |
| 3. | SK237 | August 2007 | South-eastern Arabian Sea | 26 |
| 4. | SK308 | January 2014 | Northwestern Bay of Bengal | 29 |
| 5. | SSD004 | October-November 2014 | Gulf of Mannar, Lakshadweep Sea | 41 |
| 6. | SSD055 | August 2018 | North-eastern Arabian Sea | 11 |
| 7. | SSD067 | November-December 2019 | South-western Bay of Bengal, Lakshadweep Sea, Eastern Arabian Sea | 45 |
| 8. | SSK035 | May-June 2012 | Western Bay of Bengal | 13 |
| 9. | SSK098 | January-February 2017 | Andaman Sea | 15 |

The surface sediment samples (0-1 cm) were processed following the standard procedure (Suokhrie et al., 2021b). The freeze-dried sediments were weighed and wet sieved by using 63 μm sieve. The coarse fraction (>63 μm) was dry sieved by using 250 μm and 355 μm sieves. For $\delta^{18}O$ analysis, 10-15 well preserved shells of *G. ruber* white variety were picked from 250-355 μm size range. We picked *G. ruber* s.s. wherever sufficient specimens were available. Unfortunately, several samples yielded very small carbonate fraction. In such samples, we picked mixed population of *G. ruber* to get sufficient specimens for isotopic analysis. The $\delta^{18}O_{ruber}$ was measured by using Finnigan MAT 253 isotope ratio mass spectrometer, coupled with Kiel IV automated carbonate preparation device. The samples were analyzed in the Alfred Wegner Institute for Polar and Marine Research, Bremerhaven, MARUM, University of Bremen, Bremen, Germany and the Stable Isotope Laboratory (SIL) at Indian Institute of Technology, Roorkee, India. The reference material 'National Bureau of Standards' (NBS 18/19) limestone was used as the calibration material and a secondary in-house standard was run after every 5 samples to detect and correct the drift. The precision of oxygen isotope measurements was better than 0.08‰. The $\delta^{18}O_{ruber}$ data generated on the newly collected surface sediments was augmented with the published core-top $\delta^{18}O$ measurements in the northern Indian Ocean. A total of 400 surface sediment data points (252 from this work and 148 from the previous studies) were used to understand the factors affecting $\delta^{18}O_{ruber}$ in the northern Indian Ocean (Supplementary Table 1). The annual average sea surface temperature and salinity of the top 30 m water column at the respective sample locations was downloaded from the World Ocean Atlas (Boyer et al., 2013). The salinity and temperature at the core location was extrapolated from the nearby grid points, using the Live Access Server at the National Institute of Oceanography, Goa, India.

The analyzed $\delta^{18}O_{ruber}$ data was compared with the expected $\delta^{18}O$ calcite to ascertain whether the *G. ruber* properly represents the ambient conditions. For the expected $\delta^{18}O$ calcite, the $\delta^{18}O_{sw}$ was calculated from the ambient salinity by using a compilation of the regional seawater salinity and its stable oxygen isotopic ($\delta^{18}O_{sw}$) ratio data for

the entire northern Indian Ocean (5°S to 30°N). The seawater salinity and corresponding $\delta^{18}O_{sw}$ data was downloaded
from the Schmidt et al., (1999) (version 1.22) and augmented with other regional datasets (Delaygue et al., 2001;
Singh et al., 2010; Achyuthan et al., 2013).

## 5. Results

The oxygen isotopic ratio of *G. ruber* varies from a minimum of -3.82‰ to the maximum of -1.09‰ in the surface
sediments of the northern Indian Ocean (Figure 3). The most depleted $\delta^{18}O_{ruber}$ was in the eastern BoB and the most
enriched values were in the western Arabian Sea.

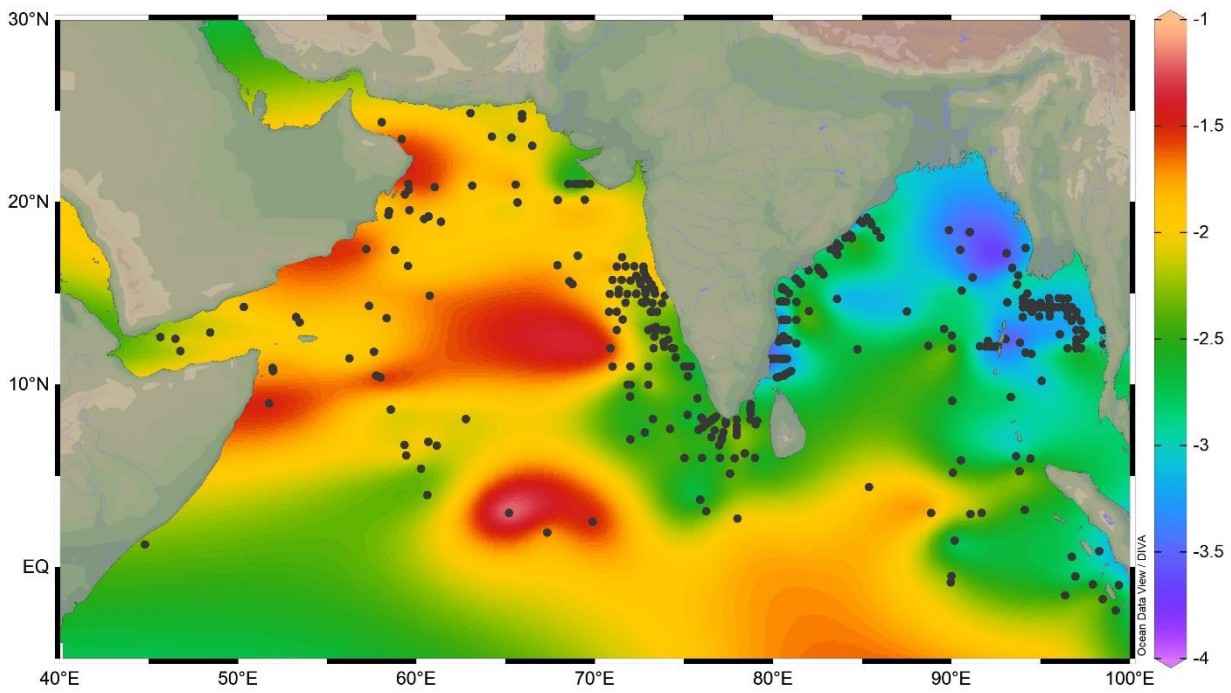

**Figure 3: The variation in *Globigerinoides ruber* $\delta^{18}O$ (‰) in the surface sediments of the northern Indian Ocean. The**
**stations are marked by black filled circle. The lowest $\delta^{18}O_{ruber}$ is in the riverine influx influenced northern Bay of Bengal**
**and the highest is in the evaporation dominated central and western Arabian Sea. The major rivers are marked with thin**
**blue lines. The map was prepared by using Ocean Data View software (Schlitzer, 2018).**

The east-west gradient in $\delta^{18}O_{ruber}$ was also evident in its significant correlation ($R^2 = 0.5$, n = 400) with the longitude
(Figure 4A). However, $\delta^{18}O_{ruber}$ did not have any systematic latitudinal variation (Figure 4B).

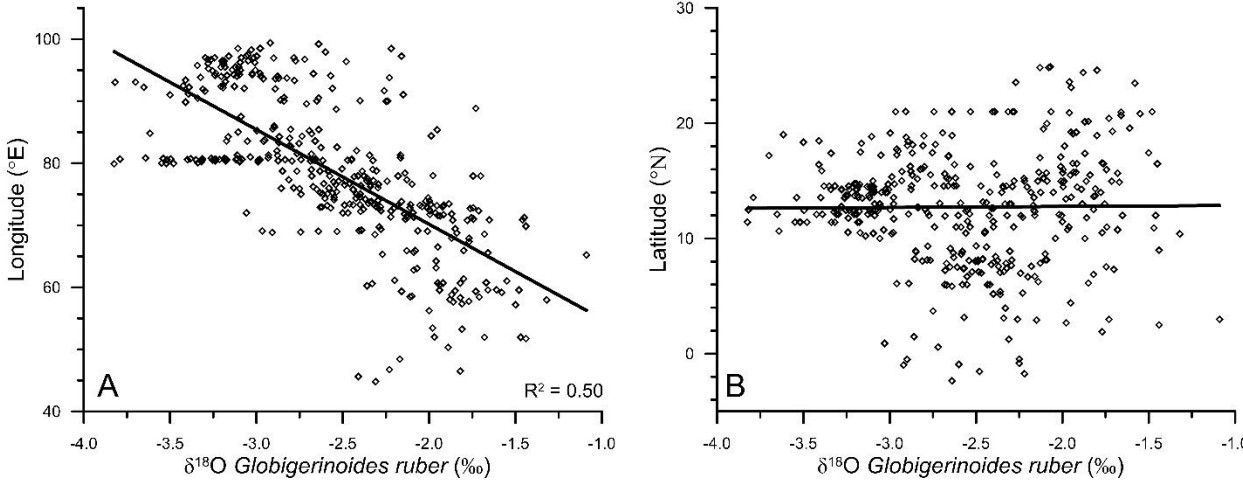

**Figure 4: The variation in *Globigerinoides ruber* δ¹⁸O (‰) with the corresponding longitude (A) and latitude (B), in the**
**surface sediments of the northern Indian Ocean. The correlation between δ¹⁸O$_{ruber}$ and latitude of the sample, is**
**insignificant.**

A significant correlation ($R^2 = 0.14$, n = 400) was observed between the water depth and δ¹⁸O$_{ruber}$ (Figure 5). δ¹⁸O$_{ruber}$
increased with increasing depth. The increase was gradual, without any abrupt change.

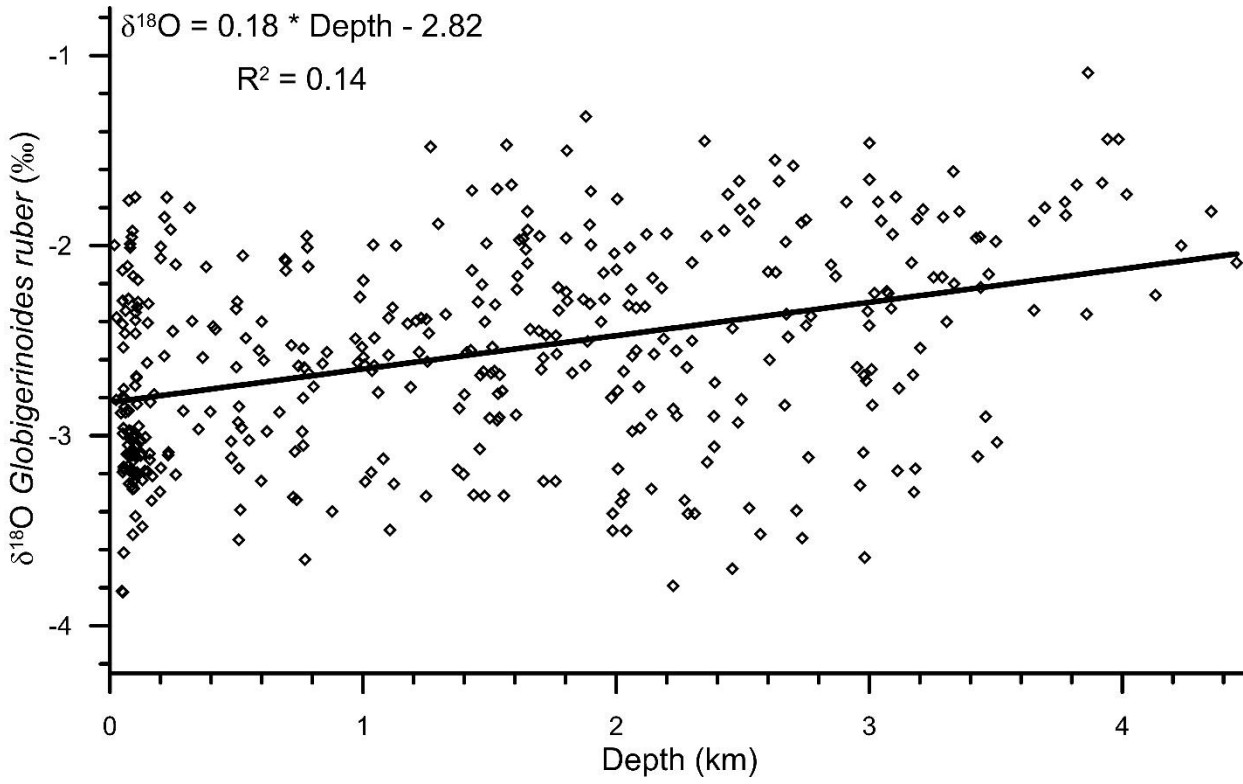

**Figure 5: The relationship between water depth and the oxygen isotopic ratio of the mixed layer dwelling *Globigerinoides***
***ruber* in the surface sediments of the northern Indian Ocean. The trendline signifies the relative enrichment of δ¹⁸O$_{ruber}$**
**shells in surface sediments, with increasing water depth. The empirical relationship between the water depth and $\delta^{18}O_{ruber}$**
**is represented with an equation. The $\delta^{18}O_{ruber}$ is significantly correlated ($R^2 = 0.14$, n = 400) with the water depth.**
The uncorrected $\delta^{18}O_{ruber}$ was significantly correlated ($R^2 = 0.63$, n = 400) with the ambient salinity (Figure 6A).
However, the relationship between uncorrected $\delta^{18}O_{ruber}$ and ambient temperature was not as robust ($R^2 = 0.18$, n =
400) (Figure 6B). A large scatter (~-3.8‰ to -1.4‰) was observed in the $\delta^{18}O_{ruber}$ of the samples collected from a
narrow range of ambient temperature (28-29°C) (Figure 6B).

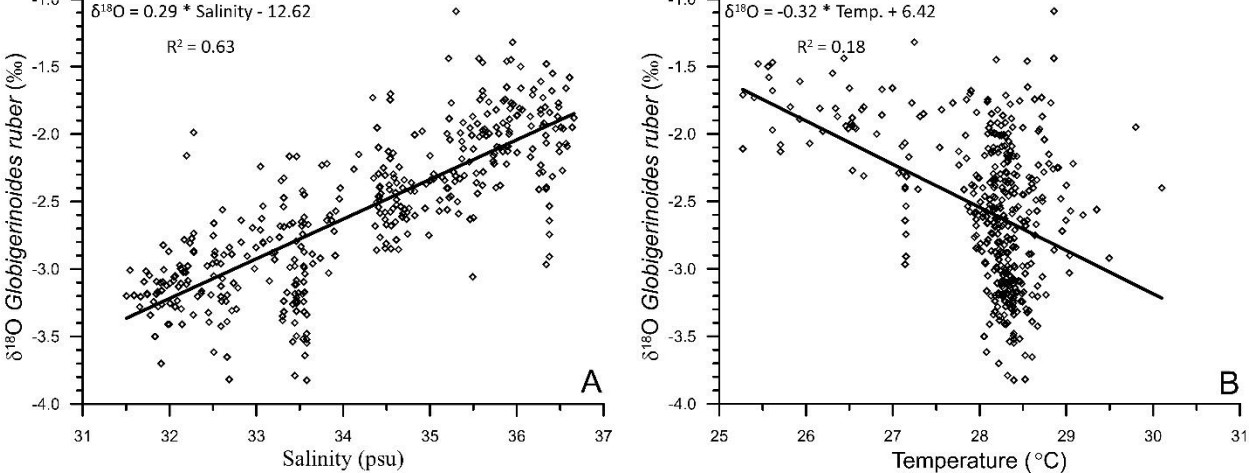

**Figure 6: The relationship between the stable oxygen isotopic ratio of the mixed layer dwelling *Globigerinoides ruber* and**
**annual average mixed layer salinity (A), and temperature (B) in the northern Indian Ocean.**
As the northern Indian Ocean includes two contrasting basins, $\delta^{18}O_{ruber}$-salinity relationship was explored for both the
Arabian Sea and the BoB. A significant $\delta^{18}O_{ruber}$-salinity relationship was observed for both the Arabian Sea ($R^2 =$
0.28, n = 205) and BoB ($R^2 = 0.14$, n = 195) (Figure 7). We report a different $\delta^{18}O_{ruber}$-salinity relationship in these
two basins. $\delta^{18}O_{ruber}$ increased with increasing salinity in both the BoB and the Arabian Sea.

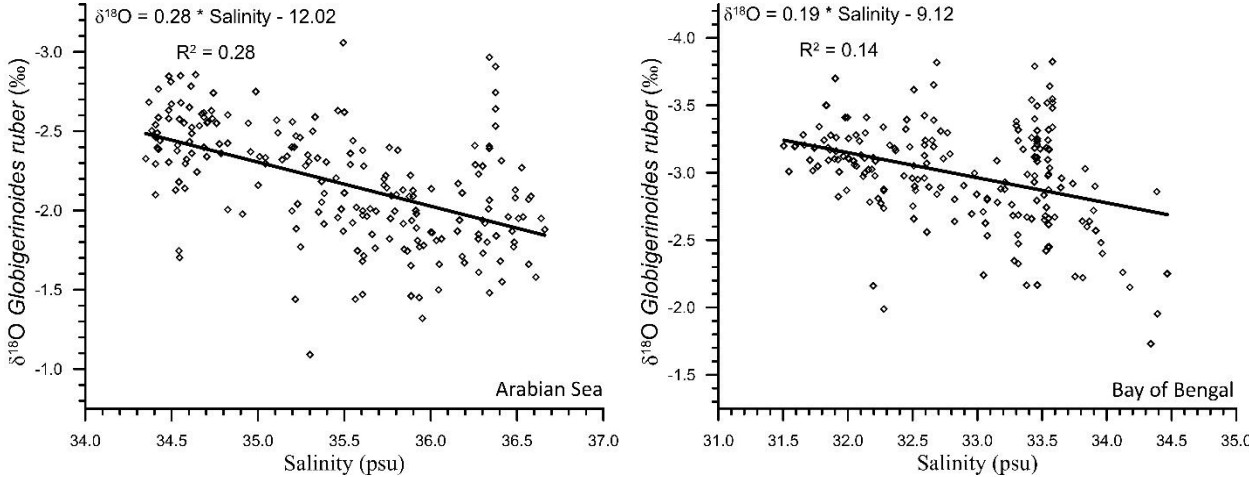


**Figure 7: The relationship between the stable oxygen isotopic ratio of mixed layer dwelling** *Globigerinoides ruber* **and annual**
**average mixed layer salinity in the Arabian Sea and Bay of Bengal.**

The dataset to derive the regional salinity-$\delta^{18}O_{sw}$ relationship comprised of a total of 750 stations with salinity varying
from 20.92 psu to 40.91 psu. The dataset also covered a large range of $\delta^{18}O_{sw}$, varying from a minimum of -2.45‰ to
the maximum of 2.02‰ (Figure 8, 9). The measured $\delta^{18}O_{ruber}$ was strongly correlated ($R^2 = 0.56$, n = 400) with the
expected $\delta^{18}O_{calcite}$, as estimated by using the salinity-$\delta^{18}O_{sw}$ relationship and the ambient temperature. However, the
relationship between seawater temperature and $\delta^{18}O_{ruber}$-$\delta^{18}O_{sw}$, was not very robust. It should however, be noted here
that the stratigraphic information is not provided for most of the core tops. The core top sediments can represent older
time slices when the sedimentation rates are low or when older sediments are exposed due to erosional processes. This
does not matter so much if the Holocene is present and stable. However, in the Indian Ocean, large Holocene $\delta^{18}O$
variations are expected due to variations in monsoon precipitation. Therefore, the uncertain age of the core tops can
affect the results stated above.

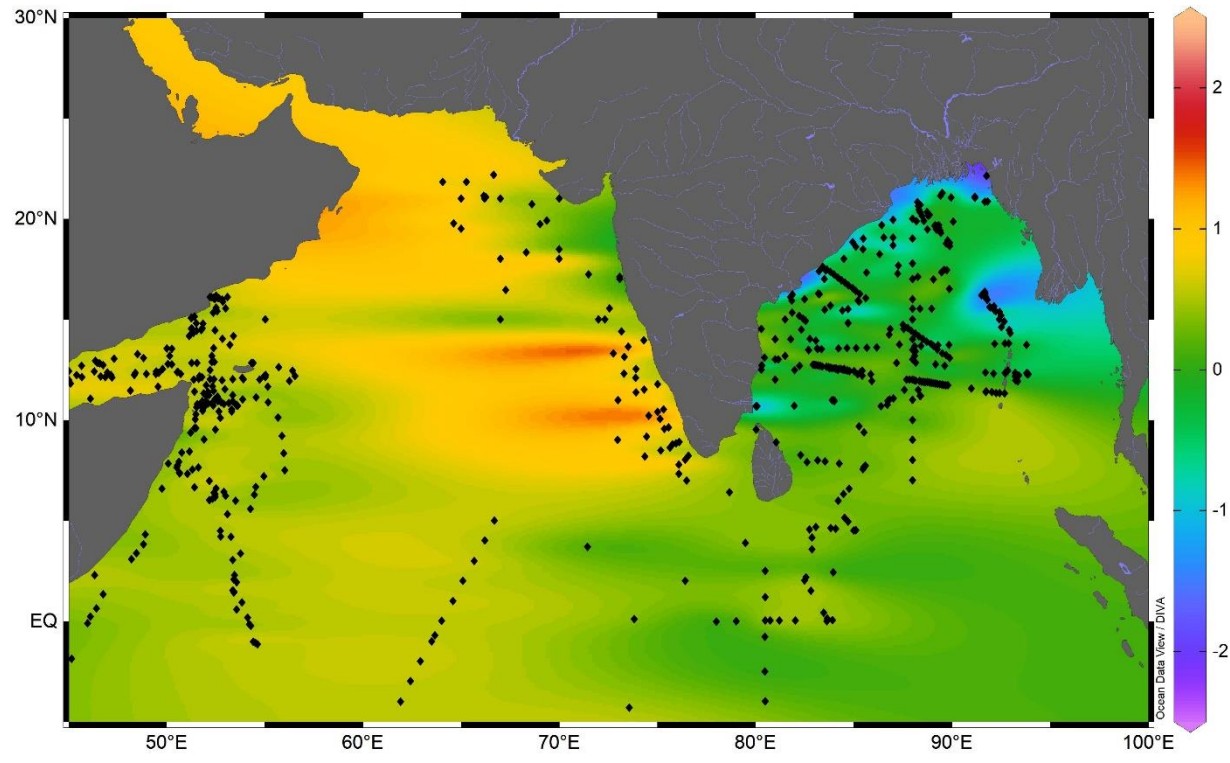




**Figure 8: The surface seawater oxygen isotopic ratio (‰) in the northern Indian Ocean. The black filled circles are the**
**seawater sample locations compiled from previous studies. The thin blue lines are the major rivers draining in the northern**
**Indian Ocean. The map was prepared by using Ocean Data View software (Schlitzer, 2018).**

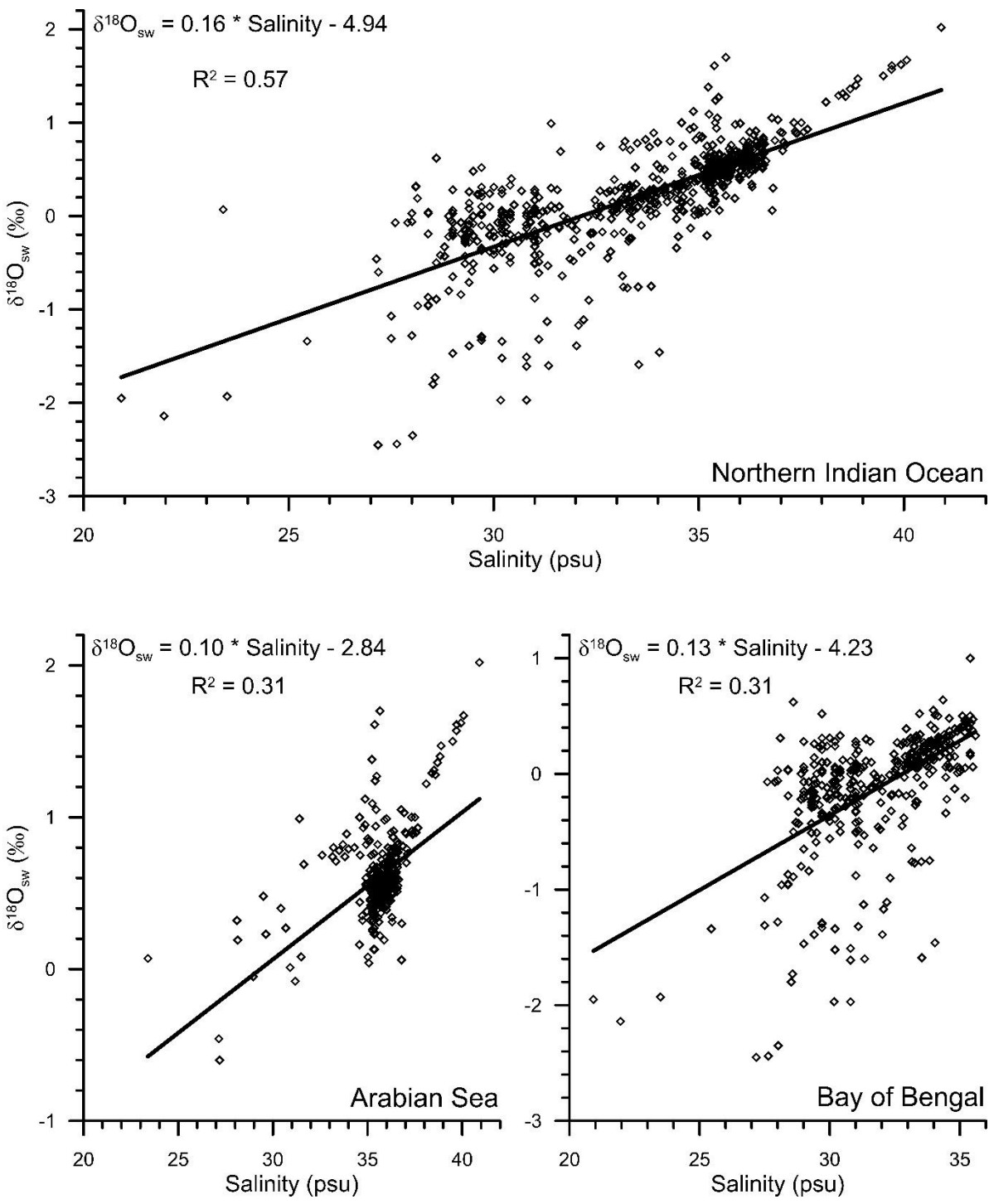

Figure 9: The relationship between surface water oxygen isotopic ratio and salinity in the northern Indian Ocean (5°S-30°N), Arabian Sea and the Bay of Bengal. The data points are from Schmidt et al., (1999), Delaygue et al., (2001), Singh et al., (2010), and Achyuthan et al., (2013).

## 6. Discussion

### 6.1 Expected versus analyzed $\delta^{18}O$

The seawater $\delta^{18}O$ data is required to estimate the expected $\delta^{18}O$ carbonate. The seawater $\delta^{18}O$, however, was not measured. Therefore, the salinity-$\delta^{18}O_{sw}$ relationship established from the previous regional seawater isotope and salinity measurements was used. The salinity-$\delta^{18}O_{sw}$ relationship varies seasonally as well as from region to region (Singh et al., 2010; Achyuthan et al., 2013; Tiwari et al., 2013). Therefore, it was difficult to choose the appropriate salinity-$\delta^{18}O_{sw}$ relationship. Initially, all the data points were clubbed to establish the salinity-$\delta^{18}O_{sw}$ relationship. By comparing the measured $\delta^{18}O_{sw}$ with the ambient salinity, we established the following relationship for the entire northern Indian Ocean (north of 5°S latitude) ($R^2 = 0.57$, n = 750) (Figure 9).

$\delta^{18}O_{sw} = 0.16*Salinity - 4.94$               Northern Indian Ocean ($R^2 = 0.57$)

Previously, a large difference in the slope of salinity-$\delta^{18}O_{sw}$ equation has been reported from the Arabian Sea and the BoB (Delaygue et al., 2001; Singh et al., 2010; Achyuthan et al., 2013). Therefore, we also plotted the salinity-$\delta^{18}O_{sw}$ separately for the Arabian Sea and BoB (Figure 9). The salinity-$\delta^{18}O_{sw}$ relationship for these two basins was represented by the following equations.

$\delta^{18}O_{sw} = 0.10*Salinity - 2.84$               Arabian Sea             ($R^2 = 0.31$, n = 375)

$\delta^{18}O_{sw} = 0.13*Salinity - 4.23$               Bay of Bengal          ($R^2 = 0.31$, n = 375)

The continuous flux of *G. ruber* throughout the year (Guptha et al., 1997) and the accumulation of shells in the sediments over a large interval, implies that the salinity-$\delta^{18}O_{sw}$ relationship based on data representing all seasons will provide a better estimate of the average $\delta^{18}O_{ruber}$ as recovered from the sediments (Vergnaud-Grazzini, 1976). The expected $\delta^{18}O_{sw}$ was calculated by using these equations and the annual average mixed layer salinity at the stations for which $\delta^{18}O_{ruber}$ data were available. The mixed layer was defined as the top 25 m of the water column following Narvekar and Prasanna Kumar (2014). Although the mixed layer depth varies regionally as well as during different seasons, the average mixed layer depth was used to compare the calcification conditions. A correction factor of 0.27‰ was applied to convert $\delta^{18}O_{sw}$ from Standard Mean Oceanic Water scale (SMOW) to Pee Dee Belemnite scale (PDB) (Hut, 1987). The expected $\delta^{18}O$ calcite was then estimated from the calculated $\delta^{18}O_{sw}$ and the annual average mixed layer temperature by using the equation proposed by Mulitza et al., (2003). We also estimated the expected $\delta^{18}O$ calcite by using the high-light equation of Bemis et al (1998), as *G. ruber* $\delta^{18}O$ is better described with the high-light equation (Thunell et al., 1999). The choice of equation used to estimate the expected $\delta^{18}O$ calcite did not make any difference other than a small offset. The difference between expected $\delta^{18}O$ calcite estimated by using paleotemperature equation of Mulitza et al., (2003) and the high-light equation of Bemis et al., (1998) varied from -0.33 to -0.41‰. The expected $\delta^{18}O$ calcite estimated by using Mulitza et al., (2003) paleotemperature equation provided values close to the

measured *G. ruber* $\delta^{18}$O. From the scatter plot (Figure 10), it was clear that the analyzed $\delta^{18}$O$_{ruber}$ was significantly
correlated ($R^2$ = 0.56, n = 400) with the expected $\delta^{18}$O calcite, suggesting that *G. ruber* closely represents the ambient
conditions in the entire northern Indian Ocean. The expected $\delta^{18}$O calcite estimated by using the separate Arabian Sea
and BoB salinity-$\delta^{18}$O$_{sw}$ equations, was also similarly correlated with the analyzed $\delta^{18}$O$_{ruber}$.

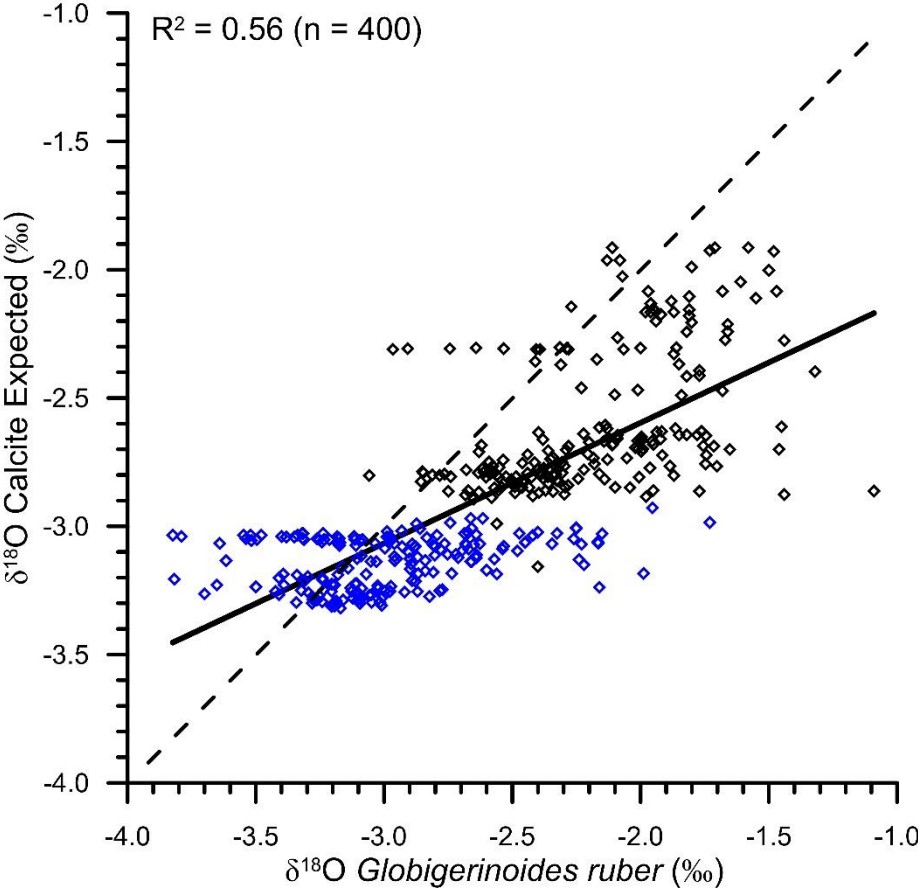

**Figure 10: The scatter plot of expected $\delta^{18}$O calcite (as estimated from the ambient salinity-temperature) and the analyzed**
**$\delta^{18}$O$_{ruber}$. The two are significantly correlated ($R^2$ = 0.56), suggesting that *Globigerinoides ruber* correctly represents the**
**ambient conditions. The blue diamonds are the samples collected from the Bay of Bengal and the black diamonds represent**
**the samples collected from the Arabian Sea. The dotted line represents the 1:1 relationship between the measured and**
**expected $\delta^{18}$O.**

The deviation of the expected $\delta^{18}$O$_{calcite}$ from the observed $\delta^{18}$O$_{ruber}$ ($\delta^{18}$O$_{residual}$) can be because of several factors
including the difference in the ambient conditions at the time of secretion of the primary calcite during the lifetime
and the digenetic changes post death and burial. The observed $\delta^{18}$O$_{ruber}$ was close to the expected $\delta^{18}$O$_{calcite}$ in the
shallower waters, especially the BoB, Andaman Sea and northeastern Arabian Sea (Figure 11). The difference was
large in the deeper Arabian Sea and the equatorial Indian Ocean. *Globigerinoides ruber* is suggested to inhabit
chlorophyll maximum for easy availability of food (Fairbanks and Weibe, 1980). In such a scenario, $\delta^{18}$O$_{ruber}$ is
expected to be higher due to lower temperatures and lower light levels at relatively deeper depths (Spero et al., 1997).
The depth of chlorophyll maximum is shallower in the marginal marine waters of both the BoB and the Arabian Sea
(Sarma & Aswanikumar, 1991; Madhu et al., 2006). If *G. ruber* thrived at chlorophyll maximum depths, the $\delta^{18}O_{ruber}$
should be enriched in heavier isotope and thus $\delta^{18}O_{residual}$ should be negative. The positive $\delta^{18}O_{residual}$ in the shallower
regions, however, suggests that *G. ruber* thrives in the warmer upper parts of the mixed layer. Alternatively, the large
influence of the depleted fresh water $\delta^{18}O$ dominates the chlorophyll maximum influence on the observed $\delta^{18}O_{ruber}$ in
the shallower regions of the northern Indian Ocean. The concentration of positive $\delta^{18}O_{residual}$ values close to the riverine
influx regions confirms the strong influence of the depleted fresh water $\delta^{18}O$ in modulating $\delta^{18}O_{ruber}$ in the northern
Indian Ocean. The negative $\delta^{18}O_{residual}$ at deeper stations is attributed to a combination of factors including deeper
chlorophyll maximum depth habitat of *G. ruber*, reduced influence of fresh water, lower sedimentation rate resulting
in mixing of older and younger fauna, and post depositional digenetic changes.

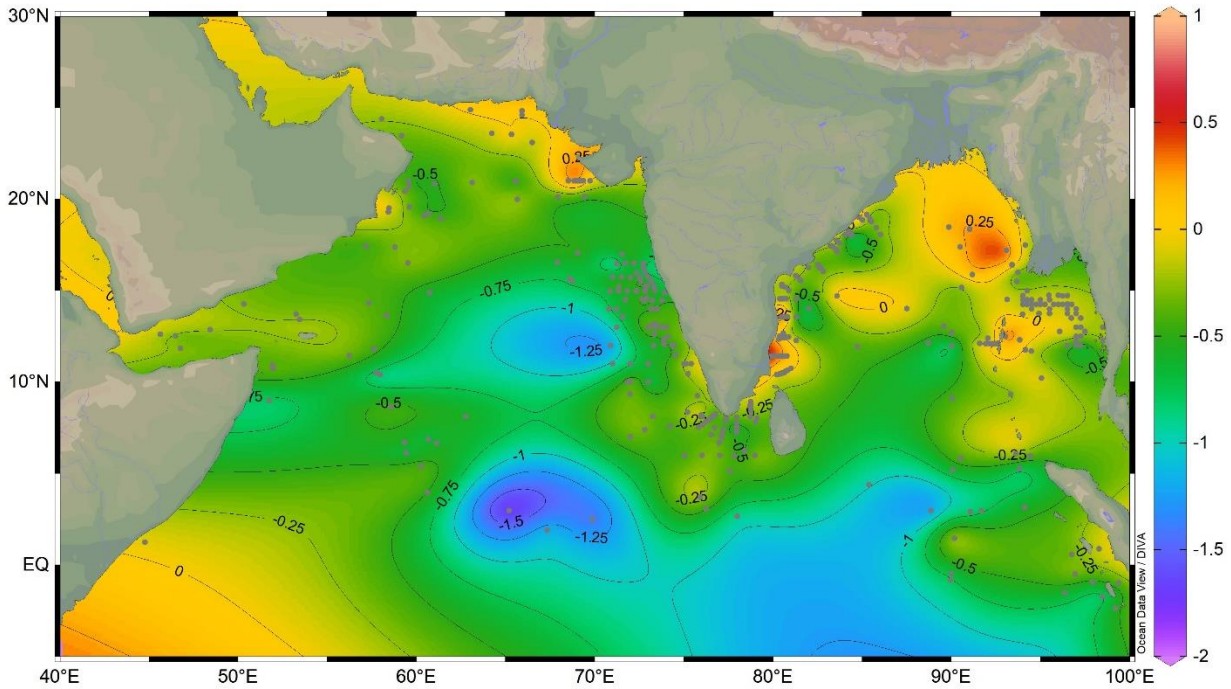


**Figure 11: The difference in the expected $\delta^{18}O_{calcite}$ and observed $\delta^{18}O_{ruber}$ ($\delta^{18}O_{residual}$) in the surface sediments of the**
**northern Indian Ocean. The grey filled squares are the sample locations. The thin blue lines are the major rivers draining**
**in the northern Indian Ocean. The thin black lines mark the contours at 0.25‰ interval. The map was prepared by using**
**Ocean Data View software (Schlitzer, 2018).**

**6.2 Latitudinal and Longitudinal variation in $\delta^{18}O_{ruber}$**
We report a strong ($R^2 = 0.50$) longitudinal influence on $\delta^{18}O_{ruber}$. A similar relationship with the latitudes is missing.
The strong longitudinal signature in $\delta^{18}O_{ruber}$ is attributed to the large salinity gradient. The huge fresh water influx in
the BoB reduces the SSS in the eastern Indian Ocean. The lack of major rivers in the western Arabian Sea results in
strong low to high salinity gradient from east to west. Although the equatorial and nearby regions are a part of the
Indo-Pacific Warm Pool, the limited temperature variability is evident in the insignificant latitudinal influence on
$\delta^{18}O_{ruber}$.

**6.3 Diagenetic alteration**
We found a strong diagenetic overprinting of $\delta^{18}O_{ruber}$ in the northern Indian Ocean (Figure 5). The enrichment of
$\delta^{18}O_{ruber}$ with increasing water depth suggests either dissolution leading to the preferential removal of chambers with
higher fraction of the lighter oxygen isotope (Wycech et al., 2018), or secondary calcification under comparatively
colder water (Lohmann, 1995; Schrag et al., 1995). The increase in planktic foraminifera $\delta^{18}O$ with increasing depth
is a common diagenetic alteration throughout the world oceans (Bonneau et al., 1980). Interestingly, the extent of the
increase in $\delta^{18}O_{ruber}$ with depth in the northern Indian Ocean is much smaller (0.18‰ per thousand meters) than that
reported for the same species from the Pacific Ocean (0.4‰ per thousand meters) (Bonneau et al., 1980). However,
the increase in $\delta^{18}O_{ruber}$ with depth in the northern Indian Ocean is continuous, unlike the abrupt shift in $\delta^{18}O$ (0.3-
0.4‰, between the depths above and below the lysocline) of another surface dwelling planktic species, namely
*Trilobatus sacculifer*, as observed in the western equatorial Pacific (Wu and Berger, 1989). The smaller increase in
$\delta^{18}O_{ruber}$ with depth is attributed to the shallower habitat of *G. ruber* as compared to *T. sacculifer*. The chamber
formation at different water depths implies increased heterogeneity in the *T. sacculifer* shells, with those formed at
warmer surface temperature being more susceptible to dissolution as compared to those formed at deeper depths during
the gametogenesis phase (Wycech et al., 2018). The chambers in *G. ruber* are formed at a similar depth and therefore,
the increase in $\delta^{18}O_{ruber}$ is continuous, while those of *T. sacculifer* are precipitated at different depths and therefore the
shift in $\delta^{18}O$ after a particular depth. The increase in $\delta^{18}O_{ruber}$ with depth is mainly due to the partial dissolution of the
more porous and thinner parts of the shells secreted at warmer temperature, as such parts are comparatively more
susceptible to dissolution (Berger, 1971). The increase in $\delta^{18}O_{ruber}$ with depth is similar in both the Arabian Sea and
BoB.

Additionally, the gradual decrease in the sedimentation rate with increasing depth and distance from the

continental margins can also induce a depth related trend in $\delta^{18}O_{ruber}$. The bioturbation disturbs the top few cm of the
sediments (Gerino et al., 1998), mixing older shells with comparatively younger shells (Löwemark, and Grootes,
2004). In high sedimentation rate regions of the shelf and slope, the mixing is restricted to the shells deposited in a
shorter, climatologically stable interval. However, in the deeper regions, it is likely that the shells deposited during
the colder glacial interval or deglaciation with relatively higher $\delta^{18}O_{ruber}$ gets mixed with the younger shells, as it is
available close to the surface due to the low sedimentation rate (Broecker, 1986; Anderson, 2001). The mixing of
shells with a relatively higher $\delta^{18}O$ with the modern shells having lighter $\delta^{18}O$ can also result in the depth related
increasing trend of $\delta^{18}O_{ruber}$. The sedimentation rate is very high on the slope and decreases in the deeper regions of
both the Arabian Sea (Singh et al., 2017) and BoB (e.g. Bhonsale and Saraswat, 2012; Suokhrie et al., 2022). However,
it should be noted here that the sedimentation rate in large parts of the BoB is still high enough to prevent availability
of deglacial or last glacial maximum shells within the bioturbation induced mixing limits.

The large influence of the terrestrial fresh water influx in the shallower region, as compared to the deeper

parts of the northern Indian Ocean, is also likely to contribute to the observed increase in $\delta^{18}O_{ruber}$ with depth. The
fresh water is depleted in heavier oxygen isotope as compared to the seawater (Bhattacharya et al., 1985; Ramesh &
Sarin, 1992). Thus, the foraminiferal shells secreted in the shallow waters are likely to be enriched in the lighter
oxygen isotope, resulting in a depth related bias. Therefore, to delineate the influence of depth related diagenetic
alteration and secondary calcification in $\delta^{18}O_{ruber}$, we subtracted the expected $\delta^{18}O_{calcite}$ from the measured $\delta^{18}O_{ruber}$.
The difference between the measured $\delta^{18}O_{ruber}$ and expected $\delta^{18}O_{calcite}$ was plotted with water depth (Figure 12). The
difference (measured $\delta^{18}O_{ruber}$ - expected $\delta^{18}O_{calcite}$) increased with depth, suggesting a strong influence of the depth
related processes in $\delta^{18}O_{ruber}$.

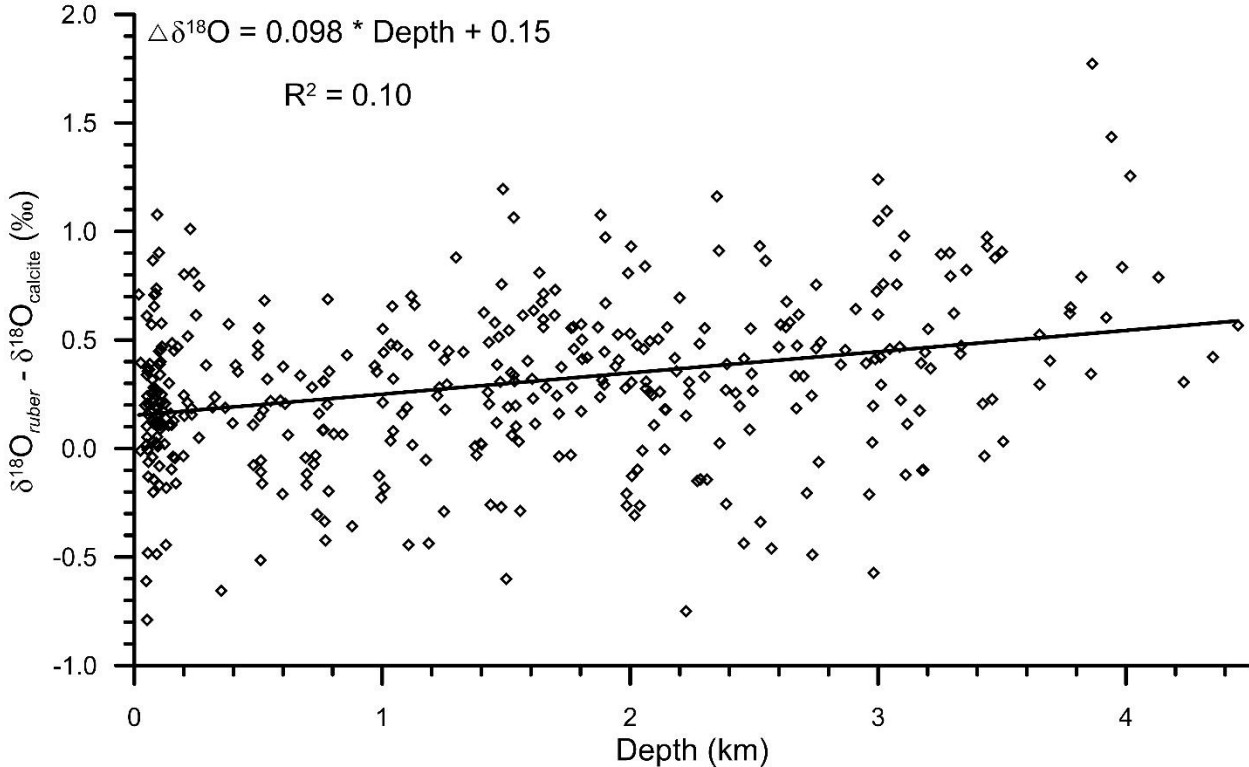


**Figure 12: The relationship of the difference between measured $\delta^{18}O_{ruber}$ and expected $\delta^{18}O_{calcite}$ with the water depth from**
**which the surface samples were collected, in the northern Indian Ocean. The $\delta^{18}O_{ruber}$ - $\delta^{18}O_{calcite}$ increased with increasing**
**water depth.**

**6.4 Salinity contribution to $\delta^{18}O_{ruber}$**
We report a strong influence of the fresh water influx induced salinity on $\delta^{18}O_{ruber}$ ($R^2$=0.63). As expected, $\delta^{18}O_{ruber}$
has a direct positive relationship with the ambient salinity. The $\delta^{18}O_{ruber}$ increased by 0.29‰ for every psu increase in
salinity. The northern Indian Ocean has a large salinity gradient (~10 psu) from the lowest in the northern BoB to the
highest in the northwestern Arabian Sea, mainly driven by the fresh water input. The river water and direct
precipitation is enriched in the lighter isotope (Kumar et al., 2010; Kathayat et al., 2021). Thus, the increased riverine
influx and precipitation contributes isotopically lighter water to the surface ocean (Rai et al., 2021) and decreases the
$\delta^{18}O_{ruber}$. From the surface seawater samples collected during the winter monsoon season (January-February 1994), a
$\delta^{18}O$-salinity slope of 0.26‰ was deduced for the Arabian Sea and of 0.18‰ for the BoB (Delaygue et al., 2001).
However, the $\delta^{18}O$-salinity slope varies regionally as well as during different seasons (Singh et al., 2010; Achyuthan
et al., 2013). The $\delta^{18}O$-salinity slope varied from as low as 0.10 for the coastal BoB samples collected during the
months of April-May to as high as 0.51 for the samples collected from the western BoB during the peak south-west
monsoon season (August-September 1988) (Singh et al., 2010). The large seasonal variation implies limitations of
$\delta^{18}O$-salinity slope deduced from snapshot surface seawater samples. Additionally, *G. ruber* flux is reported
throughout the year (Guptha et al., 1997), suggesting that the fossil population represents annual average conditions
(Thirumalai et al., 2014).
A different $\delta^{18}O_{ruber}$-salinity slope for the Arabian Sea (0.28) and BoB (0.19) is attributed to the different
hydrographic regimes of these two basins. The runoff and precipitation excess in the BoB results in a comparatively
lower salinity as compared to the evaporation dominated Arabian Sea. However, it should be noted here that the
relationship between $\delta^{18}O_{ruber}$ and salinity was very robust for all the northern Indian Ocean samples plotted together.
Interestingly, the slope of $\delta^{18}O$-salinity for the entire northern Indian Ocean samples is much lower than that for the
Atlantic Ocean (0.59 for North Atlantic and 0.52 for South Atlantic, Delaygue et al., 2000) despite the large meltwater
influx into the north Atlantic. The dissimilar $\delta^{18}O$-salinity slope in different basins and also during different seasons
in the same basin is mainly attributed to the variation in the end member composition and the relative amount of fresh
water (riverine/precipitation/sub-marine ground water discharge) input from various sources during different seasons
(Achyuthan et al., 2013; Tiwari et al., 2013). The heavier oxygen isotope depleted precipitation/fresh water influx in
the higher latitudes (~-35 ‰) as compared to the tropical areas (~-5 ‰) also results in a higher slope of the $\delta^{18}O$-
salinity relationship in the North Atlantic Ocean (Rozanski et al., 1993). Additionally, the difference in $\delta^{18}O$-salinity
slope despite of the huge fresh water input into both the basins is also because a large fraction of the riverine fresh
water spreads across the surface of the northern Indian Ocean, while the melt water sinks to deeper depths in the North
Atlantic Ocean. A consistent systematic difference has previously been observed between planktic foraminiferal shells
collected in plankton tows and surface sediments, with shells from the sediments being comparatively enriched in $^{18}O$
(Vergnaud-Grazzini, 1976).

**6.5 Temperature control on $\delta^{18}O_{ruber}$**
A first order comparison of the uncorrected $\delta^{18}O_{ruber}$ with ambient temperature of the top 30 m of the water column at
respective stations showed 0.32‰ decrease with every 1°C warming. The change in $\delta^{18}O_{ruber}$ as inferred from the
core-top sediments of the northern Indian Ocean is higher than that estimated from the plankton tows (0.22‰ per 1°C
change in temperature) (Mulitza et al., 2003). The seawater temperature was amongst the primary factors identified
to affect $\delta^{18}O_{ruber}$ (Emiliani, 1954; Mulitza et al., 2003). The low correlation between $\delta^{18}O_{ruber}$ and temperature in this
dataset is attributed to the limited temperature variability (1°C, 28-29°C) at a majority of the stations. The large salinity
difference (~6.5 psu) between stations further obscures any significant correlation between uncorrected $\delta^{18}O_{ruber}$ and
temperature. The temperature influence on $\delta^{18}O_{ruber}$ was thus assessed by comparing the ambient temperature with the
$\delta^{18}O_{ruber}$ corrected for $\delta^{18}O_{sw}$ ($\delta^{18}O_{ruber}$ - $\delta^{18}O_{sw}$). The pH of the seawater has also been identified as a factor affecting
the stable oxygen isotopic composition of planktic foraminifera (Bijma et al., 1999). However, as argued by Mulitza
et al., (2003), the limited modern surface seawater pH variability (Chakraborty et al., 2021) and its close dependence
on temperature implies that the pH contribution to $\delta^{18}O_{ruber}$ is well within the error associated with the measurements.
The seawater pH in the immediate vicinity of the foraminiferal shell is strongly influenced by the light intensity in the
presence of symbionts (Jorgensen et al., 1985). The riverine influx in the northern Indian Ocean makes the surface
waters turbid reducing the light penetration depths (Prasanna Kumar et al., 2010). Therefore, riverine influx induced
variations in turbidity in the northern Indian Ocean can influence the $\delta^{18}O_{ruber}$ via the pH effect.

The comparison of $\delta^{18}O_{sw}$ corrected $\delta^{18}O_{ruber}$ with the ambient temperature also confirms the enrichment of

($\delta^{18}O_{ruber}$ - $\delta^{18}O_{sw}$) in heavier oxygen isotope with the decrease in temperature (Figure 13). We obtained the following
relationship between temperature and ($\delta^{18}O_{ruber}$ - $\delta^{18}O_{sw}$) in the northern Indian Ocean.

Temperature = -0.59*($\delta^{18}O_{ruber}$ - $\delta^{18}O_{sw}$) + 26.40

The slope of the temperature and ($\delta^{18}O_{ruber}$ - $\delta^{18}O_{sw}$) was somewhat different (0.17‰ per 1°C change in temperature)
than that of the temperature versus uncorrected $\delta^{18}O_{ruber}$, but similar as that reported for the plankton tows (0.22‰ per
1°C change in temperature) (Mulitza et al., 2003).

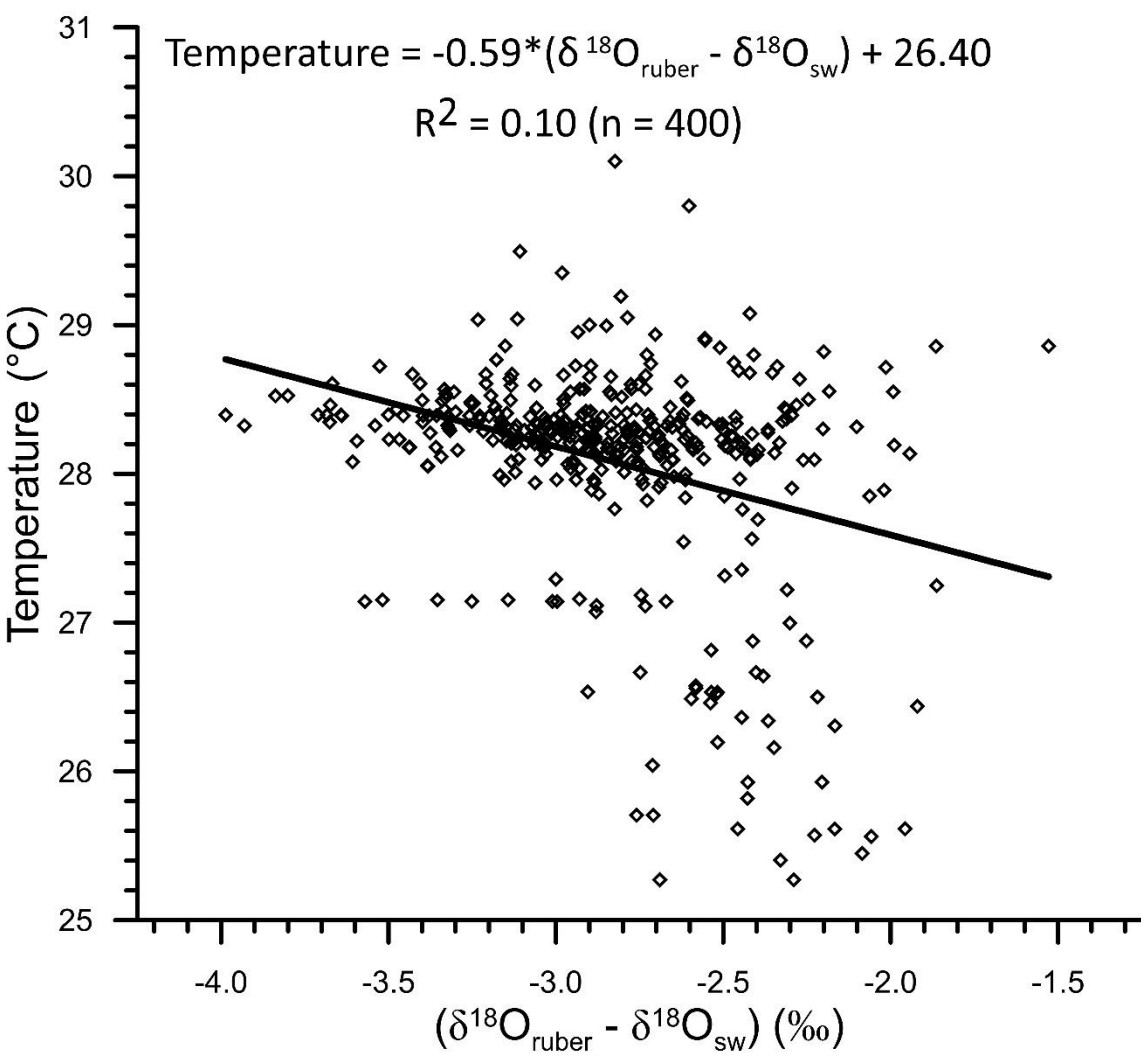

**Figure 13: The relationship between ambient temeprature and ($\delta^{18}O_{ruber}$ - $\delta^{18}O_{sw}$) for the northern Indian Ocean. As expected the ($\delta^{18}O_{ruber}$ - $\delta^{18}O_{sw}$) gets enriched in heavier isotope with decreasing ambient temperature.**

## 7. Conclusions

We measured the stable oxygen isotopic ratio of the surface dwelling planktic foraminifera *Globigernoides ruber* white variety from the surface sediments of the northern Indian Ocean. A comparison of the $\delta^{18}O_{ruber}$ with the depth suggests a strong diagenetic alteration of the isotopic ratio. The fresh water influx induced changes in the ambient salinity exert the maximum influence on the $\delta^{18}O_{ruber}$ suggesting its robust application to reconstruct the past salinity in the northern Indian Ocean. The large east-west salinity gradient in the northern Indian Ocean results in a strong longitudinal variation in $\delta^{18}O_{ruber}$. The temperature influence on $\delta^{18}O_{ruber}$ is subdued as compared to the effect of large salinity variation in the northern Indian Ocean. We report a relatively smaller change in $\delta^{18}O_{ruber}$ with a unit increase

in ambient temperature in case of specimens retrieved from the surface sediments as compared to those collected live
from the water column.
**8. Data Availability**
The newly generated data as well as the data compiled from previous studies from the northern Indian Ocean has been
submitted to PANGAEA and is available at
https://www.pangaea.de/tok/59190adf9e4facf7ebb9ad555c0bce58a9a72bd9 (Saraswat et al., 2022). The data is
submitted with the manuscript as well, for the reviewers' scrutiny.
**9. Author Contribution**
RS designed the research, compiled and interpreted the data and wrote the manuscript. TS, DKN, DPS, SMS, MS,
GS, SRB, SRK picked the specimens for isotopic analysis. MM, ASM supervised the analysis. All authors edited and
contributed to the final manuscript.
**10. Competing Interests**
The authors declare that they have no conflict of interest.
**Acknowledgements**
We thank the crew onboard expeditions during which the surface sediment samples were collected. The authors thank
the Director, CSIR-National Institute of Oceanography for the facilities and funding. The technical personnel at the
Alfred-Wegner Institute for Polar and Marine Research, and MARUM, Bremen University, Germany, are
acknowledged for the help in stable isotopic analysis. We thank Dr. P.S. Rao and Dr. V. Ramaswamy, CSIR-NIO for
providing the surface sediment samples collected from the Myanmar continental shelf. The authors also thank Dr.
B.N. Nath for providing the spade core-top samples from the eastern margin of India. We thank Dr. Alberto Sanchez,
Centro Interdisciplinario de Ciencias Marinas, Instituto Politécnico Nacional, La Paz, B.C.S, Mexico, and the
anonymous reviewer, for their constructive comments and suggestions that helped to improve the manuscript.

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
