# Peer review of "Large fresh water influx induced salinity gradient and diagenetic changes in the northern Indian Ocean dominate the stable oxygen isotopic variation in *Globigerinoides ruber"

_Earth System Science Data, 2022_

## Referee Comment (RC2)

[referee-annotated manuscript omitted]

---

## Author Comment (AC1)

**Authors' Response to Reviewer's Comments**

**Reviewer's Comment:** Saraswat et al. present an extensive core-top compilation of new and published δ18O values of the species Globigerinoides ruber. G. ruber is the most commonly measured planktic species and the new data set has the potential to improve the coverage in the Indian Ocean and can complement existing global compilations (i.e., Waelbroeck et al. 2005). The data set itself would benefit from additional documentation (data source, sampling depth, δ13C of G. ruber) that can be added with minimal effort. As outlined below, the scientific analysis and interpretation of the data need revision:

**Authors' Response:** Thank you very much for appreciating our effort. We have added the details (data source, sampling depth), wherever available. As the $\delta^{13}C$ of *G. ruber* is missing for a majority of the previous data points, we have not included it, at this point. We have incorporated all the suggestions regarding the scientific analysis and interpretation and hope the revised manuscript will be accepted for publication.

**Reviewer's Comment:** Title/L18, 20, 23, 25, 34…This is probably only a wording issue but "salinity" is not influencing or controlling the δ18O of carbonate, δ18O of sea water is. The authors probably use "salinity" because salinity is often reconstructed from δ18O of seawater and because both parameters are locally highly correlated and influenced by precipitation and evaporation. However, I think the authors should be precise. Salinity and δ18O are related but can be decoupled to some extent, i.e., if the source of the freshwater endmember changes, the same salinity might be associated with different δ18O seawater values. This is demonstrated by the large scatter of the the δ18O-salinity relationships in Figure 8 and the different freshwater endmembers in the Arabian Sea (-2.84 ‰) and the BOB (-4.23 ‰).

**Authors' Response:** We agree with the reviewer that $\delta^{18}O$ carbonate depends on $\delta^{18}O$ sea water. However, as pointed out by the reviewer, huge riverine influx and direct precipitation strongly influence the salinity as well as the $\delta^{18}O$ sea water in the northern Indian Ocean. In view of the strong influence of the fresh water influx on both the salinity and $\delta^{18}O$ sea water, $\delta^{18}O$ carbonate is often used to reconstruct the salinity changes in the northern Indian Ocean. It was to highlight this close relationship between fresh water influx induced salinity and $\delta^{18}O$ carbonate, that we stated it in the title. As the reviewer has rightly suggested that the change in endmember $\delta^{18}O$ can affect the $\delta^{18}O$ carbonate without a concomitant change in salinity, we have rephrased the title as given below.
'Large fresh water influx induced salinity gradient and diagenetic changes in the northern Indian Ocean dominate the stable oxygen isotopic variation in *Globigerinoides ruber*'

**Reviewer's Comment:** L39: There are earlier examples of δ18O seawater reconstructions based on G. ruber that should be cited here including Wang et al. (1995), Kallel et al. (1997) and Schmidt et al. (2004).

**Authors' Response:** The suggested references have been added.

**Reviewer's Comment:** L44: I disagee with this statement. The relationships between the physicochemical conditions and foraminiferal δ18O seems to be quite constant since we have a very good agreement between laboratory derived equations and field data when the ambient conditions of shell formation are known. The reasons why there are offsets in different basins is that, particularly

when core tops are used, the exact conditions of shell formation (δ18O sea water, light intensity, temperature, depth of calcification, seasonal flux, pH…) are not well constrained.

**Authors' Response:** The sentence has been deleted.

**Reviewer's Comment:** L113/Methods: I suggest to show a table with all the relevant information, the expeditions, number of samples, region etc.

**Authors' Response:** The following table with all the details, has been added.

| Sr.No. | Cruise | Month/Year | Area | Total Samples |
|--------|--------|------------|------|---------------|
| 1. | SK117 | September-October 1996 | Eastern Arabian Sea | 27 |
| 2. | SK175 | April-May 2002 | North-eastern Bay of Bengal | 45 |
| 3. | SK237 | August 2007 | South-eastern Arabian Sea | 26 |
| 4. | SK308 | January 2014 | Northwestern Bay of Bengal | 29 |
| 5. | SSD004 | October-November 2014 | Gulf of Mannar, Lakshadweep Sea | 41 |
| 6. | SSD055 | August 2018 | North-eastern Arabian Sea | 11 |
| 7. | SSD067 | November-December 2019 | South-western Bay of Bengal, Lakshadweep Sea, Eastern Arabian Sea | 45 |
| 8. | SSK035 | May-June 2012 | Western Bay of Bengal | 13 |
| 9. | SSK098 | January-February 2017 | Andaman Sea | 15 |

**Reviewer's Comment:** L131: There seems to be a small discrepancy between the values obtained from published sources and the original data. According to PANGAEA and the appendix in the cited thesis by Sirocko (1989), the core top value for MD76-135 is -1.81 ‰ (this paper: -1.89 ‰) and for MD76-136 -1.78 ‰ (this paper: -1.82 ‰). Perhaps this offset is just due to a different sampling depth or averaging. I have not checked the other sites. Please check and document the sampling depth in the core and the actual source of the data (table in paper, supplement or data base with doi/link)
MD76-136: https://doi.org/10.1594/PANGAEA.77485
MD76-135: https://doi.org/10.1594/PANGAEA.77484
Sirocko (1989): https://doi.org/10.2312/reports-gpi.1989.27

**Authors' Response:** We have used MARGO compilation (Waelbroeck et al., 2005, https://doi.pangaea.de/10.1594/PANGAEA.227321) for the previously published northern Indian Ocean core-top/Late Holocene *G. ruber* $\delta^{18}$O data. We checked the individual core data available on PANGAEA and observed small difference in the values due to different core depths, as stated by the reviewer. The values have been corrected for the top 2 cm of the core, wherever the data is available (e.g. MD76-135, -1.89 replaced with -1.81).

**Reviewer's Comment:** Figure 1, 2, 7: The maps have been produced with OceanDataView. Please give proper credit to the author of the software and cite Schlitzer, R., Ocean Data View, https://odv.awi.de, 2018. What kind of interpolation technique has been used to produce the maps?

**Authors' Response:** The following sentence has been added in all the maps and reference has been added in the list.

'The map has been prepared by using Ocean Data View software (Schlitzer, 2018).'

**Reviewer's Comment:** L146: State the laboratory where the samples have been measured.

**Authors' Response:** The following details have been added.

'The samples were analyzed in the Alfred Wegner Institute for Polar and Marine Research, Bremerhaven, MARUM, University of Bremen, Bremen, Germany and the Stable Isotope Laboratory (SIL) at Indian Institute of Technology, Roorkee, India.'

**Reviewer's Comment:** Figure 4: The relation of δ18O with water depth is interesting. To what extend is this relation influenced by the fact that shallow sites are more exposed to freshwater input (=lower δ18O) than the more remote deeper sites? To isolate the effect of water depth, it might help to plot the difference to the predicted δ18O vs water depth rather than absolute δ18O.

**Authors' Response:** Thank you very much for the suggestion. We have added a plot of the difference between the measured $\delta^{18}O_{ruber}$ and expected $\delta^{18}O_{calcite}$.

[Figure]

**Figure 11: The relationship of the difference between measured $\delta^{18}O_{ruber}$ and expected $\delta^{18}O_{calcite}$ with the water depth from which the surface samples were collected, in the northern Indian Ocean. The $\delta^{18}O_{ruber}$ - $\delta^{18}O_{calcite}$ increases with increasing water depth.**

The following text has also been added.

'The large influence of the terrestrial fresh water influx in the shallower region, as compared to the deeper parts of the northern Indian Ocean, is also likely to contribute to the observed increase in $\delta^{18}O_{ruber}$ with depth. The fresh water is depleted in heavier oxygen isotope as compared to the seawater (Bhattacharya et al., 1985; Ramesh & Sarin, 1992). Thus, the foraminiferal shells secreted in the shallow waters are likely to be enriched in the lighter oxygen isotope, resulting in a depth related bias. Therefore, to delineate the influence of depth related diagenetic alteration and secondary calcification in $\delta^{18}O_{ruber}$, we subtracted the expected $\delta^{18}O_{calcite}$ from the measured $\delta^{18}O_{ruber}$. The difference between the measured $\delta^{18}O_{ruber}$ and expected $\delta^{18}O_{calcite}$ was plotted with water depth (Figure 11). The difference (measured $\delta^{18}O_{ruber}$ - expected $\delta^{18}O_{calcite}$) increased with depth, suggesting a strong influence of the depth related processes in $\delta^{18}O_{ruber}$.'

**Reviewer's Comment:** Figure 5 & 6: What is the purpose of showing separate relations with temperature and salinity (must be δ18O of sea water, see above) when we know that both factors influence the δ18O of foraminiferal carbonate? Temperature and salinity are also spatially related and it is impossible to quantify the relative influence of both factors in the plots. In my opinion, Figures 5 and 6 and the corresponding discussion can be omitted.

**Authors' Response:** We agree with the reviewer that $\delta^{18}O_{sw}$ depends on both the salinity and temperature. However, we hope the reviewer will agree with us that the relative influence of salinity and temperature on $\delta^{18}O_{sw}$, will depend on the extent of variation in these parameters. A large range of seawater salinity is observed in the northern Indian Ocean, as compared to the limited variation in temperature. Also, many researchers have used the salinity estimated from the residual $\delta^{18}O_{sw}$ to reconstruct past salinity and in turn monsoon intensity. In view of the difference in the range of variation in the salinity and temperature, we plotted the $\delta^{18}O$ separately, to understand the relative influence of these two parameters. Therefore, we have retained the figure and discussion. However, if the reviewer insists, we'll remove this section.

**Reviewer's Comment:** Figure 9: I suggest to have the same ranges/limits for x and y axis and to indicate the 1:1 relation with an additional line. Is the slope of the shown regression statistically significantly different from 1? To assess whether the derived linear equations will help to improve paleoclimatic reconstructions (as stated in L27), it would be important to quantify the error of the regression equations.

**Authors' Response:** As advised, same range has been used for both the x and y axes. We have also added a line representing 1:1 relationship between the measured and expected $\delta^{18}O$.

[Figure]

**Figure 9: The scatter plot of expected δ$^{18}$O calcite as estimated from the ambient salinity-temperature and the analyzed δ$^{18}$O$_{ruber}$. The two are significantly correlated (R$^2$ = 0.56), suggesting that *Globigerinoides ruber* correctly represents the ambient conditions. The dotted line represents the 1:1 relationship between the measured and expected δ$^{18}$O.**

**Reviewer's Comment:** L155: Salinity from the World Ocean Atlas is provided on a grid. How exactly was salinity estimated at the core location? From the nearest grid point?

**Authors' Response:** Yes, the salinity and temperature at the core location was extrapolated from the nearby grid points, using the Live Access Server at the National Institute of Oceanography, Goa, India. The details have been added in the manuscript.

**Reviewer's Comment:** L218/Discussion: It should be mentioned that stratigraphic information is not provided for most of the core tops. Core top sediments can represent older time slices when the sedimentation rates are low or when older sediments are exposed due to erosional processes. This does not matter so much if the Holocene is present and stable. However, in the Indian Ocean large Holocene δ18O variations are expected due to variations in Monsoon precipitation.

**Authors' Response:** The following details have been added.

'It should however, be noted here that the stratigraphic information is not provided for most of the core tops. Core top sediments can represent older time slices when the sedimentation rates are low or when older sediments are exposed due to erosional processes. This does not matter so much if the Holocene is present and stable. However, in the Indian Ocean large Holocene δ$^{18}$O variations are expected due to variations in monsoon precipitation. Therefore, the uncertain age of the core tops can affect the results stated above.'

**Reviewer's Comment:** L 245: Explain how mixed layer depth has been defined/determined?

**Authors' Response:** The following details have been added.

'The mixed layer was defined as the top 25 m of the water column following Narvekar and Prasanna Kumar (2014). Although the mixed layer depth varies regionally as well as during different seasons, the average mixed layer depth was used to compare the calcification conditions.'

**Reviewer's Comment:** L246: Provide more context. Why has the O. universa low light equation been used if G. ruber δ18O is better described with the high-light equation (see Thunell et al. 1999)? How big are the constant offsets to the published equations?

**Authors' Response:** The following details have been added.

'We also estimated the expected δ$^{18}$O calcite by using the high-light equation of Bemis et al (1998), as *G. ruber* δ$^{18}$O is better described with the high-light equation (Thunell et al. 1999). The choice of equation used to estimate the expected δ$^{18}$O calcite did not make any difference other than a small offset. The difference between expected δ$^{18}$O calcite estimated using paleotemperature equation of Mulitza et al., (2003) and the high-light equation of Bemis et al., (1998) varied from -0.33 to -0.41‰. The expected δ$^{18}$O calcite estimated using Mulitza et al., (2003) paleotemperature equation provided values close to the measured *G. ruber* δ$^{18}$O.'

**Reviewer's Comment:** L247/Discussion/Figure 9: It would be very interesting to see a map of the residual δ18O (predicted-observed). Any spatial pattern in the residuals might point to the environmental/ecological reason for the deviations from expected δ18O. For example, the chlorophyll

maximum is very shallow in the coastal areas of the BOB and the Arabian Sea whereas it is deep in the central Indian ocean. If G. ruber exploits the chlorophyl maximum as a food source (Fairbanks and Wiebe, 1980) it might record higher δ18O due to lower temperatures and lower light levels (i.e., Spero et al., 1997). Likewise, the deeper central areas should show lower sedimentation rates and stronger signals from dissolution and sediment mixing.

**Authors' Response:** We gained a lot of insight after plotting the map of expected-observed $\delta^{18}O$. We have discussed the same in the revised manuscript and added the following text and figure.

The deviation of the expected $\delta^{18}O_{calcite}$ from the observed $\delta^{18}O_{ruber}$ ($\delta^{18}O_{residual}$) can be because of several factors including the difference in the ambient conditions at the time of secretion of the primary calcite during the lifetime and the digenetic changes post death and burial. The observed $\delta^{18}O_{ruber}$ was close to the expected $\delta^{18}O_{calcite}$ in the shallower waters, especially the Bay of Bengal, Andaman Sea and northeastern Arabian Sea (Figure 10). The difference was large in the deeper Arabian Sea and the equatorial Indian Ocean. *Globigerinoides ruber* is suggested to inhabit chlorophyll maximum for easy availability of food (Fairbanks and Weibe, 1980). In such a scenario, $\delta^{18}O_{ruber}$ is expected to be higher due to lower temperatures and lower light levels at relatively deeper depths (Spero et al., 1997). The depth of chlorophyll maximum is shallower in the marginal marine waters of both the Bay of Bengal and the Arabian Sea (Sarma & Aswanikumar, 1991; Madhu et al., 2006). If *G. ruber* thrived at chlorophyll maximum depths, the $\delta^{18}O_{ruber}$ should be enriched in heavier isotope and thus $\delta^{18}O_{residual}$ should be negative. The positive $\delta^{18}O_{residual}$ in the shallower regions, however, suggests that *G. ruber* thrives in the warmer upper parts of the mixed layer. Alternatively, the large influence of the depleted fresh water $\delta^{18}O$ dominates the chlorophyll maximum influence on the observed $\delta^{18}O_{ruber}$ in the shallower regions of the northern Indian Ocean. The concentration of positive $\delta^{18}O_{residual}$ values close to the riverine influx regions confirms the strong influence of depleted fresh water $\delta^{18}O$ in modulating $\delta^{18}O_{ruber}$ in the northern Indian Ocean. The negative $\delta^{18}O_{residual}$ at deeper stations is attributed to a combination of factors including deeper chlorophyll maximum depth habitat of *G. ruber*, reduced influence of fresh water, lower sedimentation rate resulting in mixing of older and younger fauna, and post depositional digenetic changes.'

[Figure]

**Figure 10: The difference in the expected $\delta^{18}O_{calcite}$ and observed $\delta^{18}O_{ruber}$ in the surface sediments of the northern Indian Ocean. The grey filled squares are the sample locations. The thin blue lines are the major rivers draining in the northern Indian Ocean. The thin black lines mark the contours at 0.25‰ interval. The map has been prepared by using Ocean Data View software (Schlitzer, 2018).**

**Reviewer's Comment:** L270. The authors attribute the increase of δ18O with water depth to diagenesis and partial dissolution. This is certainly a possible explanation. However, it seems very likely that the deep basins receive much less terrigenous matter than the shallow slope/shelf sites and that the sedimentation rates are significantly lower in the deep parts of the Indian Ocean. When the sedimentation rates are lower than ~2 cm/kyr (quite common in deep pelagic basins) it is likely that the the Holocene is mixed with isotopically heavier material from the last glacial/deglaciation (Broecker 1986), which might also explain the relationship of δ18O with water depth. Is anything known about the sedimentation rates in the deep basins?

**Authors' Response:** Thank you very much for pointing out this aspect. We have added the following details.

'Additionally, the gradual decrease in the sedimentation rate with increasing depth and distance from the continental margins can also cause a depth related trend in $\delta^{18}O_{ruber}$. The bioturbation disturbs the top few cm sediments (Gerino et al., 1998) resulting in the mixing of older shells with comparatively younger shells (Löwemark, and Grootes, 2004). In high sedimentation rate regions of the shelf and slope, the mixing is restricted to the shells deposited in a shorter climatologically stable interval. However, in the deeper regions, it is likely that the shells deposited during the colder glacial interval or deglaciation with relatively higher $\delta^{18}O_{ruber}$ gets mixed with the younger shells, as it is available close to the surface due to the low sedimentation rate (Broecker, 1986; Anderson, 2001). The mixing of shells with a relatively higher $\delta^{18}O$ with the modern shells having lighter $\delta^{18}O$ can also result in the depth related increasing trend of $\delta^{18}O_{ruber}$. The sedimentation rate is very high on the slope and decreases in the deeper regions of both the Arabian Sea (Singh et al., 2017) and Bay of Bengal (e.g. Bhonsale and Saraswat, 2012; Suokhrie et al., 2022).'

**Reviewer's Comment:** L291-293: Unfortunately, δ13C of G. ruber has not been not reported and shown along with δ18O. As DIC in river water is very depleted in carbon-13, since it derives from the remineralized organic matter, it provides additional information on freshwater input (see for example Pastouret et al. 1978). If freshwater from river input is the reason for some of the low δ18O values in G. ruber in the BOB, δ13C values should be low as well. Since δ18O and δ13C of foraminiferal carbonate are always measured together it should be straightforward to add δ13C of G. ruber to the data set.

**Authors' Response:** We agree that the fresh water influx also affects the carbon isotopic ratio of the foraminiferal shells. However, a larger variation in the carbon isotopic ratio in the northern Indian Ocean is due to the large productivity changes. The productivity is invariably high in the marginal marine regions as compared to the open ocean, mainly due to the upwelling and nutrient input. Therefore, it is difficult to delineate the relative contribution of the fresh water influx and productivity in the carbon isotopic ratio of the foraminifera in the northern Indian Ocean. Additionally, the $\delta^{13}C$ of a majority of the previously published records is not available. Therefore, it is difficult to add the $\delta^{13}C$ data.

**Reviewer's Comment:** L306-313: Please review and revise the reasons for the different slopes in the North Atlantic and the Indian Ocean. The higher slope of the δ18O/salinity relationship in the North Atlantic is due to the fact that precipitation/freshwater in high latitudes is depleted in oxygen-18 (~-35 ‰) compared to tropical areas (~-5 ‰) (see for example Figure 10 in Rozanski et al., 1993).

**Authors' Response:** We have added the following lines.

'The heavier oxygen isotope depleted precipitation/fresh water influx in the higher latitudes (~-35 ‰) as compared to the tropical areas (~-5 ‰) also results in a higher slope of the $\delta^{18}$O-salinity relationship in the North Atlantic Ocean (Rozanski et al., 1993).'

**Reviewer's Comment:** L324: Except for upwelling regions, the variations in surface water pH are indeed small. However, the pH at the foraminiferal shell is strongly influenced by light intensity in the presence of symbionts (see Jorgensen et al. 1985, Fig 6). Small variations in turbidity or calcification depth can therefore influence the stable isotope ratio of G. ruber via the pH effect.

**Authors' Response:** We have added the following text in the revised manuscript.

'The seawater pH in the immediate vicinity of the foraminiferal shell is strongly influenced by the light intensity in the presence of symbionts (Jorgensen et al., 1985). The riverine influx in the northern Indian Ocean makes the surface waters turbid reducing the light penetration depths (Prasanna Kumar et al., 2010). Therefore, riverine influx induced variations in turbidity in the northern Indian Ocean can influence the $\delta^{18}$O$_{ruber}$ via the pH effect.'

---

## Author Comment (AC2)

**Authors' Response to Reviewer's Comments**

**Reviewer's Comment:** Title. Salinity influences d18Osw and indirectly d18Oruber. The title should be more specific.
**Authors' Response:** We agree with that the salinity indirectly influences the oxygen isotopic ratio of planktic foraminifera. The title has been changed to
'Large fresh water influx induced salinity gradient and diagenetic changes in the northern Indian Ocean dominate the stable oxygen isotopic variation in *Globigerinoides ruber*'

**Reviewer's Comment:** Introduction. The distinguishment of s.s. and s.l. morphotypes in G. ruber is crucial for the subject of the paper dealing with. which one(s) was (were) used? or was it a mixture.
**Authors' Response:** The following details have been added.
'We picked *G. ruber* s.s. wherever sufficient specimens were available. Unfortunately, several samples yielded very small carbonate fraction. In such samples, we picked mixed population of *G. ruber* to get sufficient specimens for isotopic analysis.'

**Reviewer's Comment:** Include other factors that impact in the d18Oruber, for example, photosynthesis, symbionts, pH, alkalinity, among others.
**Authors' Response:** The following text has been added in the introduction.
'The ambient seawater pH, carbonate ion concentration (Bijma et al., 1999), presence/absence of symbionts (Jørgensen et al., 1985) also affect the isotopic composition of *G. ruber*. However, limited glacial-interglacial variability in these parameters is masked by the dominance of temperature and fresh water influx induced salinity changes in oxygen isotopic ratio of *G. ruber.*'

**Reviewer's Comment:** How does the oxygen isotopic composition of G. ruber change with shell dissolution? Explain.
**Authors' Response:** The following sentence has been added.
'The dissolution preferentially removes depleted $\delta^{18}O$ sections of the shells, thus increasing the whole shell $\delta^{18}O_{ruber}$ (Berger and Gardner, 1975; Lohmann, 1995; Weinkauf et al., 2020).'

**Reviewer's Comment:** Similar studies have been conducted in the Pacific and Atlantic oceans and require citation, for example:
Farmer, E. C., A. Kaplan, P. B. de Menocal, and J. Lynch-Stieglitz (2007), Corroborating ecological depth preferences of planktonic foraminifera in the tropical Atlantic with the stable oxygen isotope ratios of core top specimens, Paleoceanography, 22, PA3205, doi:10.1029/2006PA001361.
Steph, S., Regenberg, M., Tiedemann, R., Mulitza, S., & Nürnberg, D. (2009). Stable isotopes of planktonic foraminifera from tropical Atlantic/Caribbean core-tops: Implications for reconstructing upper ocean stratification. rine Micropaleontology, 71(1-2), 1-19.
Thirumalai, K., Richey, J.N., Quinn, T.M. & Poore, R.Z. Globigerinoides ruber morphotypes in the Gulf ofMexico: A test of null hypothesis. Sci. Rep. 4, 6018; DOI:10.1038/srep06018 (2014).
Sánchez, A., Sánchez-Vargas, L., Balart, E., & Domínguez-Samalea, Y. (2022). Stable oxygen isotopes in planktonic foraminifera from surface sediments in the California Current system. Marine Micropaleontology, 173, 102127.

**Authors' Response:** All these studies have been referred in the introduction. The following text has been added.

'The $\delta^{18}O$ of surface dwelling planktic foraminifera *Globigerinoides ruber* ($\delta^{18}O_{ruber}$) is often used to reconstruct past surface seawater conditions (Saraswat et al., 2012; 2013; Mahesh and Banakar, 2014). Therefore, continuous efforts are made to understand the factors affecting $\delta^{18}O_{ruber}$ (Vergnaud-Grazzini, 1976; Multiza et al., 1997; 2003; Waelbroeck et al., 2005; Mohtadi et al., 2011; Horikawa et al, 2015; Hollstein et al., 2017; ; Sanchez et al., 2022). The depth habitat of *G. ruber* in the tropical Atlantic Ocean has been inferred from its stable oxygen isotopic ratio (Farmer et al., 2007). The change in stable oxygen isotopic ratio of planktic foraminifera, including *G. ruber*, is suggested as a proxy to reconstruct upper water column stratification in the tropical Atlantic Ocean, based on the good correlation between $\delta^{18}O$ and the ambient seawater characteristics (Steph et al., 2009). A few studies suggested a difference in the $\delta^{18}O$ of various morphotypes of *G. ruber* (sensu stricto and sensu lato) and attributed it to their distinct ecology and depth habitat (Löwemark et al., 2005). However, a recent study from the Gulf of Mexico suggested a similar ecology and depth habitat for both the *G. ruber* morphotypes (Thirumalai et al., 2014).'

**Reviewer's Comment:** Materials and Methodology. what certified or secondary reference standards were used?.

**Authors' Response:** The reference material NBS 18 limestone was used as the calibration material and a secondary in-house standard was run after every 5 samples to detect and correct the drift. The details have been added in the methodology.

**Reviewer's Comment:** In which laboratories each group of samples were measured.

**Authors' Response:** The following details have been added.

'The $\delta^{18}O_{ruber}$ was measured by using Finnigan MAT 253 isotope ratio mass spectrometer, coupled with Kiel IV automated carbonate preparation device. The samples were analyzed in the Alfred Wegner Institute for Polar and Marine Research, Bremerhaven, MARUM, University of Bremen, Bremen, Germany and the Stable Isotope Laboratory (SIL) at Indian Institute of Technology, Roorkee, India.'

**Reviewer's Comment:** Provides a table specifying the date, number of samples, type of instrument used for collection and other relevant information for each cruise, including literature data.

**Authors' Response:** The following details have been added for the newly analysed samples.

| Sr.No. | Cruise | Month/Year | Area | Total Samples |
|--------|--------|------------|------|---------------|
| 1. | SK117 | September-October 1996 | Eastern Arabian Sea | 27 |
| 2. | SK175 | April-May 2002 | North-eastern Bay of Bengal | 45 |
| 3. | SK237 | August 2007 | South-eastern Arabian Sea | 26 |
| 4. | SK308 | January 2014 | Northwestern Bay of Bengal | 29 |
| 5. | SSD004 | October-November 2014 | Gulf of Mannar, Lakshadweep Sea | 41 |
| 6. | SSD055 | August 2018 | North-eastern Arabian Sea | 11 |
| 7. | SSD067 | November-December 2019 | South-western Bay of Bengal, Lakshadweep Sea, Eastern Arabian Sea | 45 |
| 8. | SSK035 | May-June 2012 | Western Bay of Bengal | 13 |

| 9. | SSK098 | January-February 2017 | Andaman Sea | | 15 |

**Reviewer's Comment:** Include the distribution of salinity and temperature under contrasting conditions with and without freshwater input.

**Authors' Response:** We have included the following figure with sea surface temperature and sea surface salinity during the non-monsoon months of April-May and summer monsoon months of August-September.

[Figure]

**Figure 1: The sea surface temperature (SST) (°C) (Locarnini et al., 2018) and salinity (SSS) (psu) (Zweng et al., 2018) in the northern Indian Ocean during the monsoon (August-September) and non-monsoon (April-May)**

**months. Major rivers draining into the northern Indian Ocean, are marked by blue lines. The map has been prepared by using Ocean Data View software (Schlitzer, 2018).**

**Reviewer's Comment:** Results. Include the P value in the correlation analysis.
**Authors' Response:** The correlation value ($R2 = 0.5$, n = 400) has been added.

**Reviewer's Comment:** Figure 9: Use color symbols to differentiate the d18O values of Arabian Sea and BoB. Why is R2 different in the figure with respect to the figure caption?.
**Authors' Response:** The figure has been revised to show the samples from the Bay of Bengal and Arabian Sea in different coloured symbols. The $R^2$ was wrongly mentioned in caption. It has been revised now.

[Figure]

**Figure 9: The scatter plot of expected $\delta^{18}O$ calcite as estimated from the ambient salinity-temperature and the analyzed $\delta^{18}O_{ruber}$. The two are significantly correlated ($R^2 = 0.56$), suggesting that *Globigerinoides ruber* correctly represents the ambient conditions. The blue diamonds are the samples collected from the Bay of Bengal and the black diamonds represent the samples collected from the Arabian Sea. The dotted line represents the 1:1 relationship between the measured and expected $\delta^{18}O$.**

**Reviewer's Comment:** Fig. 4: The coefficient of determination (R2) is very small, what other parameters are involved in this relationship?. Note that the isotopic change is larger over the same depth interval with respect to the 0.18 change. Why do you avoid describing this change in the d18Oruber?
**Authors' Response:** The oxygen isotopic ratio is mainly affected by the ambient seawater $\delta^{18}O$ (which in turn depends on fresh water influx), temperature and pH. The digenetic changes and secondary

calcification also affects the $\delta^{18}O_{ruber}$. Here, we wanted to see the effect of depth related digenetic processes on $\delta^{18}O_{ruber}$. The correlation is low because depth related processes exert significant but very low control on $\delta^{18}O_{ruber}$ as compared to the effect of fresh water influx induced change in ambient seawater $\delta^{18}O$ and temperature, as discussed in further sections.

**Reviewer's Comment:** Fig. 5B: Describe in results and explain in discussion, the large variability in d18Oruber values with respect to the temperature range of 28 to 29oC.
**Authors' Response:** The following sentence has been added in the results.
'A large scatter (~-3.8‰ to -1.4‰) was observed in the $\delta^{18}O_{ruber}$ of the samples collected from a narrow range of ambient temperature (28-29°C).'

The following details are already mentioned in the discussion.
'The low correlation between $\delta^{18}O_{ruber}$ and temperature in this dataset is attributed to the limited temperature variability (1°C, 28-29°C) at a majority of the stations. The large salinity difference (~6.5 psu) between stations further obscures any significant correlation between uncorrected $\delta^{18}O_{ruber}$ and temperature.'

**Reviewer's Comment:** Include a Fig. of the spatial distribution of d18O of calcite in isotopic equilibrium.
**Authors' Response:** In response to a comment by other reviewer, we have included a figure of the spatial distribution of equilibrium isotopic composition minus the observed $\delta^{18}O_{ruber}$. Based on that figure, the possible factors responsible for the shift of $\delta^{18}O_{ruber}$ from the equilibrium isotopic composition are discussed. We believe that figure addresses the point raised by the reviewer. If not, then we will add another figure of the equilibrium isotopic composition.

**Reviewer's Comment:** Discussion. The discussion is very descriptive. The discussion needs improvement. So I recommend the authors to review the recent literature. Some aspects of interest are distinguishing between s.s. and s.l. morphotypes and how it affects oxygen isotopic composition, depth of calcification of G. ruber in the Indian Ocean and other oceans, symbionts and chlorophyll, carbonate ion, depth of lysocline i.e. dissolution, shell size. All these aspects will enhance the discussion.
**Authors' Response:** The discussion has been thoroughly revised in view of the comments by both the reviewers.

**Reviewer's Comment:** Please also note the supplement to this comment:
**Authors' Response:** The comments/suggestions in the annotated manuscript have been addressed and included in this rebuttal.